# Ribosome biogenesis in plants requires the nuclear envelope and mitochondria localized OPENER complex

Wei Wang [1,4] ✉, Amir Mahboubi [2], Shaochun Zhu [3], Johannes Hanson [2], André Mateus [3] & Totte Niittylä [1] ✉

Eukaryotic ribosome biogenesis proceeds from nucleolus to cytosol assisted by various assembly factors. The process is evolutionarily conserved across eukaryotes but differences between the kingdoms are emerging. Here, we describe how the OPENER (OPNR) protein complex is required for 60S ribosome assembly in the model plant *Arabidopsis thaliana*. The complex is observed on both nuclear envelope and mitochondria, and contains OPNR, OPENER ASSOCIATED PROTEIN 1 (OAP1), OAP2, Cell Division Cycle 48 D (CDC48D) and Calmodulin-interacting protein 111 (CIP111). Depletion of the OPNR complex components results in reproductive lethality and cytoplasmic retention of assembly factors on 60S ribosomes. Subsequent biochemical analyses and structural modelling suggest that OPNR, OAP1 and OAP2 form a claw-like trimer which grabs the ribosome assembly factor RIBOSOMAL PROTEIN L24C (RPL24C) on the pre-60S ribosome. Our results reveal previously unrecognised subcellular complexity of ribosome biogenesis in plants, and point to mitochondria association as a feature to ensure sufficient translational capacity.

Ribosomes are large RNA and protein complexes that all living cells need for protein biosynthesis. In eukaryotes, four ribosomal RNAs (rRNAs) and ~80 proteins form the mature cytoplasmic 80S ribosome, which comprises a 60S large subunit and a 40S small subunit. Our understanding of ribosome biogenesis is mainly based on yeast and animal cell studies, but sequence similarity based comparative studies indicate that the process is largely conserved also in plants[1]. However, sequence similarity-based approaches are likely to miss evolutionarily distant kingdom and organism specific mechanisms. In all eukaryotes the 60S and 40S subunits are assembled in separate multistep processes which ultimately converge in the cytosol to form the mature ribosomes. Ribosome biogenesis is an energy-demanding cellular process which requires careful regulation[2]. This regulation is governed by tens of small nucleolar RNAs (snoRNAs) and hundreds of ribosomal biogenesis factors, each of which has distinct roles in rRNA folding, processing, modification, ribosomal protein binding, ribosome quality control and subunit transport[3,4].

Ribosome subunit biogenesis starts in the nucleolus with the production of ribosomal RNAs (rRNAs) and the incorporation of ribosomal proteins. The subunit precursors are then moved from the nucleolus to the nucleoplasm, where additional proteins are incorporated before the pre-60S and pre-40S ribosomes are exported through the nuclear pore into the cytoplasm for the assembly of the mature 60S and 40S subunits. A critical part of the ribosome subunit assembly is carried out by placeholder proteins, which temporarily bind to specific sites in the pre-ribosome subunits until they are replaced by the final site-specific proteins. This way placeholders prevent the premature incorporation of ribosome components and function as molecular clocks to control the intricate assembly process[5].

[1]Umeå Plant Science Centre, Department of Forest Genetics and Plant Physiology, Swedish University of Agricultural Sciences, 90183 Umeå, Sweden. [2]Umeå Plant Science Centre, Department of Plant Physiology, Umeå University, 90183 Umeå, Sweden. [3]Department of Chemistry, Umeå University, 90736 Umeå, Sweden. [4]Present address: Yazhouwan National Laboratory, Sanya 572025 Hainan, China. ✉e-mail: wangwei@yzwlab.cn; totte.niittyla@slu.se

A key placeholder during the assembly of the 60S subunit is the evolutionarily conserved Ribosomal-like protein 24 (Rlp24). First identified in yeast, the Rlp24 is incorporated into the pre-60S ribosome in the nucleolus, functioning as a placeholder until it is replaced by a paralog, large subunit r-protein L24 (Rpl24), in the cytoplasm[5,6]. Rlp24 and Rpl24 have distinct C-terminal domains, but share similar N-terminal domains which are incorporated into the 60S subunit. In yeast, the replacement of the Rlp24 with Rpl24 is carried out by a specific type II AAA+ (ATPases Associated with diverse cellular Activities) ATPase called Diazaborine resistance gene 1 (Drg1) also known as the ATPase family gene 2 (AFG2). Drg1 contains two tandem AAA+ ATPase domains (D1 and D2 domains) linked by a short linker (D1-D2 linker), and an amino-terminal domain (N-domain) which interacts with Rlp24[6,7]. Structural studies have revealed that Drg1 forms a homohexamer with a double-ring structure and a central tunnel[6,7]. Following pre-60S ribosome export through the nuclear pore, the N-domain of Drg1 binds to the C-terminus of Rlp24 to extract it from the pre-60S ribosome. The interaction between Rlp24 and Drg1 stimulates the ATPase activity of Drg1, which then catalyzes Rlp24 release[8-12]. Drg1 also interacts with the phenylalanine-glycine (FG) repeats of nucleoporins located on the cytosolic surface of the nuclear pore complex. This interaction between Drg1 and the FG repeats in nucleoporin Nup116 promotes the release of Rlp24, which suggests a coupling between the export of pre-60S ribosomes and the initiation of cytoplasmic maturation in yeast[10].

Structural studies of yeast pre-60S ribosomal particles have shown that Rlp24 also interacts with two other maturation factors called Nucleolar GTP-binding protein 1 (Nog1) and Bud20[13,14]. More specifically, the release of Rlp24 is a prerequisite for the subsequent release of Nog1 and Bud20 during the final steps of pre-60S ribosomal maturation[11,12]. Nog1 is a small GTPase involved in pre-60S ribosome export from the nucleus and subsequent maturation in the cytosol[15,16]. Bud20 is a rRNA-binding protein featuring a zinc finger motif that is also required for the nuclear export of the pre-60S subunit. Following release from the pre-60S ribosome in the cytoplasm, Bud20 is recycled back into the nucleus[17].

Plant ribosome biogenesis shares certain parallels with the process that has been described for yeast, yet also exhibits features presumably reflecting unique processes in plant growth and development. Interestingly, some ribosome biogenesis proteins also appear to be absent in plants or show low sequence similarity to their yeast and animal counterparts[1,18]. In this study, we show that the Arabidopsis gene *OPENER* (*OPNR*) plays a critical role in ribosome biogenesis. We originally identified *OPNR* in a search for essential, evolutionarily conserved single-copy *Arabidopsis* genes of unknown function, and showed it to be essential for seed development and meristem maintenance[19]. Prior experiments have demonstrated that *opnr* mutants exhibit enlarged nucleoli and smaller mitochondria, and that the OPNR protein shows an unusual localization on both nuclear envelope and mitochondria in plant cells[19]. To uncover the true function of OPNR we performed co-immunoprecipitation (co-IP) coupled with mass spectrometry analyses to identify OPNR-associated proteins[19], and discovered that OPNR, OAP1, OAP2 form a heterotrimer. We provide evidence that this trimer functions as an adapter of a AAA+ ATPase complex which is required for 60S ribosome subunit biogenesis. Our findings identify previously unknown features of ribosome biogenesis in plants, and show that although the core components appear conserved across eukaryotes some critical details differ.

## Results

### Mutations in *OPENER ASSOCIATED PROTEIN 1* (*OAP1*) and *OAP2* causes similar seed abortion phenotype as *opnr*

Our previously published co-IP results revealed two uncharacterized proteins which lacked known functional domains among the highest ranked interactors of OPNR[19]. These proteins were named OPENER

ASSOCIATED PROTEIN 1 (OAP1, AT5G07950), and OPENER ASSOCIATED PROTEIN 2 (OAP2, AT3G49645). To verify that these proteins genuinely interact with OPNR, we first investigated the phenotypes of the corresponding mutants. For *OAP1* a T-DNA insertion line was obtained from the Nottingham Arabidopsis Stock Centre (Nottingham, UK). For *OAP2*, one mutant line with a 216 bp gene body deletion was created by CRISPR/Cas9 since no T-DNA insertion lines were available (Supplementary Fig. 1). In order to isolate the deletion carrying line it was necessary to cross the CRISPR/Cas9 expressing transgenic lines with wild-type Col-0. This allowed the isolation of *oap2* mutant lines without an active CRISPR/Cas9 cassette. As was the case with *opnr*, no homozygous *oap1* and *oap2* mutants could be isolated and the heterozygous *oap1*/+ and *oap2*/+ siliques showed an early ovule/seed abortion phenotype (Fig. 1A). The *oap1*/+ ovule/seed abortion phenotype was rescued by the expression of *OAP1-mNeonGreen* fusion protein under the control of the *OAP1* promoter (*pOAP1:OAP1-mNeonGreen*). Moreover, the *oap2*/+ phenotype was rescued by the expression of wild-type *OAP2* gene driven by the *OAP2* native promoter (*pOAP2:OAP2*) (Fig. 1A). These results confirmed that the early ovule/seed abortion phenotype observed in both *oap1*/+ and *oap2*/+ mutants was caused by the mutations of *OAP1* and *OAP2*, respectively.

To investigate the development of ovules and seeds in the *oap1*/+ and *oap2*/+ mutants in detail, unpollinated ovules and young seeds representing wild-type and mutant plants were cleared in Hoyer's solution[20]. The ovules and seeds were observed under a differential interference contrast (DIC) microscopy where nucleoli are more prominently visible relative to the nuclei[20]. Similar to the phenotype of *opnr*, the nucleoli of egg cells and synergids in the *oap1*/+ and *oap2*/+ female gametophyte were clearly larger than what was observed for wild-type samples (Fig. 1B). In both *oap1* and *oap2* mutants, embryo development was arrested at the zygote stage, with the mutants demonstrating clearly larger nucleoli at the zygote stage when compared to wild-type samples. Furthermore, apical vacuoles, which are a characteristic feature of *opnr*, were also observed in arrested zygotes of both *oap1* and *oap2* (Fig. 1C). These results indicate that OPNR, OAP1 and OAP2 are functionally related and essential proteins for ovule and seed development.

### Two OPNR-associated AAA+ ATPases are required for gametophyte development and plant reproduction

The proteins OPNR, OAP1, and OAP2 contain no obvious functional domains which would be indicative of a specific molecular function. We reasoned that further investigation of the proteins identified via OPNR co-IP experiments could elucidate the functional role of these essential proteins. The top-ranked proteins identified via co-IP included CDC48D and CIP111, both of which belong to the large family of AAA+ ATPases[19,21,22]. The genes that encode for these proteins were selected for mutant analysis.

A 466-bp deletion in *CDC48D* gene body was obtained by CRISPR/Cas9 (Supplementary Fig. 1C). The self-pollinated *cdc48D*/+ mutant plants showed reduced seed set, with 48.7% of the ovules/seeds ($n = 511$) aborting at an early stage of development; the corresponding ratio for wild-type siliques was 2.8% ($n = 471$) abortions (Fig. 1A). The pollination of *cdc48D*/+ pistils with wild-type pollen resulted in the abortion of 49.5% ($n = 499$) of ovules/seeds in the resulting siliques, which mirrored the self-pollination results (Fig. 1A). These results demonstrate that *cdc48D* mutation influences ovule development. Closer investigation of the ovules revealed that nearly half of the ovules had developmental defects (20 out of 43). Minority of these abnormal ovules (four out of 20) had arrested at different early developmental stages (Fig. 1B)[23], while the majority (16 out of 20) had reached stage FG7[23], yet showed enlarged nucleoli when compared to wild-type ovules (Fig. 1B). Expression of the *pCDC48D:mScarlet-CDC48D* fusion protein under the control of the *CDC48D* native promoter was able to rescue the *cdc48D* mutant ovule abortion phenotype; more specifically, the transformants showed a

very low abortion rate (1.1%, *n* = 266) (Fig. 1A). Hence, the obtained results revealed that CDC48D is essential for ovule development in Arabidopsis.

The CRISPR/Cas9 system was also used to create *CIP111* mutants. Heterozygous *cip111* deletion mutants were identified in the primary transformants. Initially, these *cip111* CRISPR/Cas9 lines were crossed

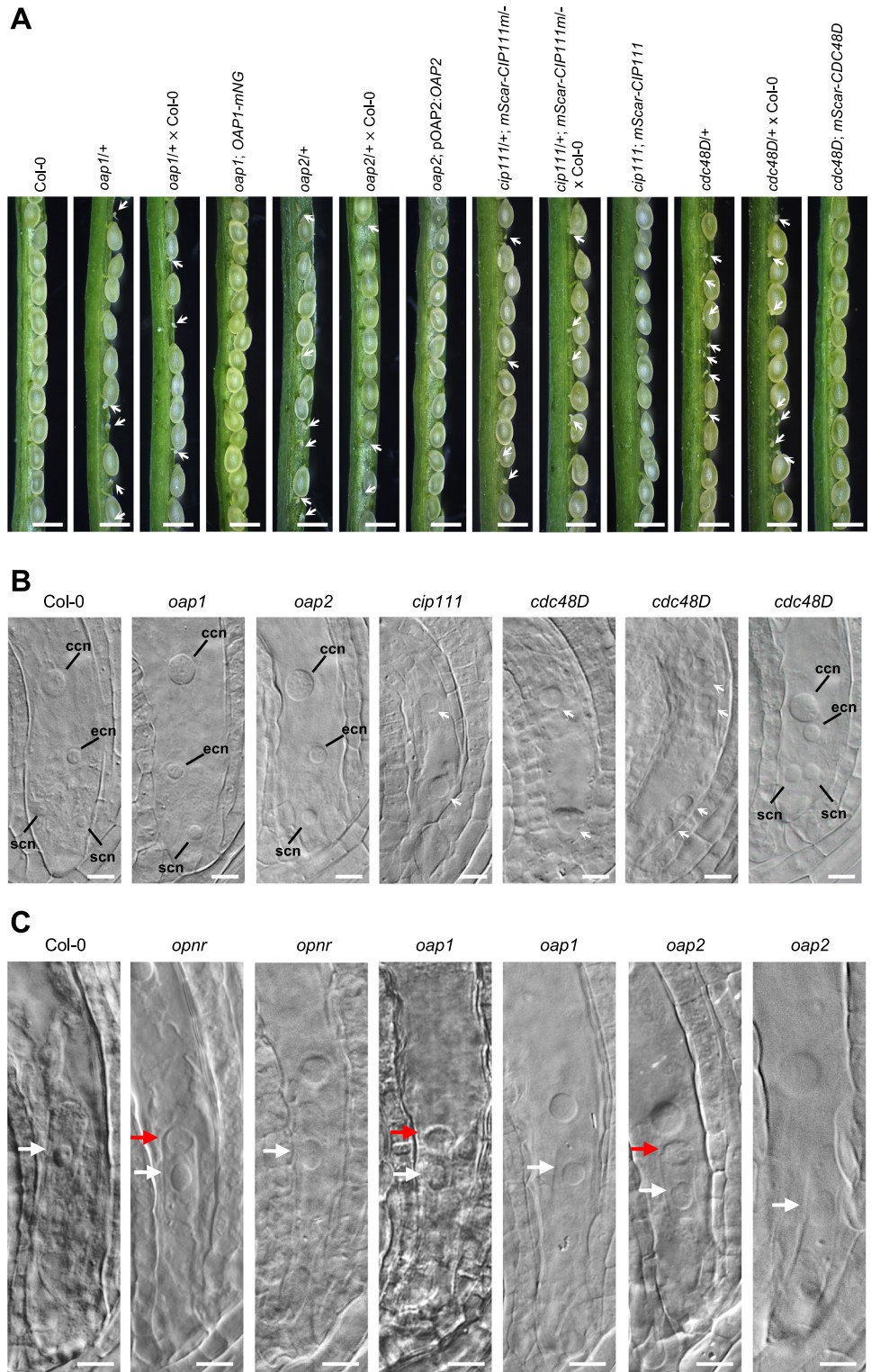

**Fig. 1 | Phenotypes of *OAP1*, *OAP2*, *CDC48D* and *CIP111* mutants. A** Developing seeds in siliques collected from Col-0 control plants, *oap1*, *oap2*, *cdc48D* and *cip111* mutants, along with the complemented mutants and mutants crossed with Col-0 plants. White arrows indicate aborted ovules or seeds. *ap1; OAP1-mNG* is *oap1* mutant complemented with *pOAP1:OAP1-mNeonGreen*. *oap2; pOAP2:OAP2* is *oap2* mutant complemented with *pOAP2:OAP2*. *cip111; mScar-CIP111m* is *cip111* mutant complemented with *pCIP111:mScarlet-CIP111m*. *cdc48D; mScar-CDC48D* is *cdc48D* mutant complemented with *pCDC48D:mScarlet-CDC48D*. **B** Ovules from Col-0 control plants, *oap1*, *oap2*, *cdc48D*, and *cip111* mutant plants. White arrows indicate the nucleolus. ccn: central cell nucleolus, ecn: egg cell nucleolus, scn: synergid cell nucleolus. **C** Zygotes from Col-0 control plants, *opnr*, *oap1* and *oap2* mutant plants. Red arrows indicate the vacuole, while white arrows indicate the nucleolus. Scale bars: **A** 0.5 mm; **B**, **C** 10 μm.

with Col-0, but no *cip111* lines without the CRISPR/Cas9 cassette could be identified in the offspring (*n* = 50). This observation suggested that the *cip111* mutation cannot be transmitted to the next generation through a male or female gametophyte. In order to circumvent the suspected lethality of *cip111* mutation, we created a mutated *CIP111* gene with modified CRISPR/Cas9 targeting site (*CIP111m*) that does not change the amino acid sequence. Expression of the modified *pCIP111:mScarlet-CIP111m* fusion protein, which was under the control of the native promoter, in the *cip111* CRISPR/Cas9 background enabled the identification of a 583-bp deletion in *CIP111* (Supplementary Fig. 1D). This line was crossed with wild-type plants to obtain homozygous *cip111/cip111; mScarlet-CIP111m/mScarlet-CIP111m* lines without the CRISPR/Cas9 transgene. The siliques of these plants exhibited an abortion frequency that was close to what was observed for wild-type plants (1.3%, *n* = 388, Fig. 1A). We also obtained heterozygous *cip111/+* mutants with heterozygous complementation transgene *pCIP111:mScarlet-CIP111m/-* (*cip111/+; mScarlet-CIP111m/-*); these mutants are expected to produce male and female gametophytes which harbor a *cip111* mutation without the presence of the complementation transgene. The *cip111/+; mScarlet-CIP111m/-* siliques showed a 26.8% ovule/seed abortion rate (*n* = 776), and when female plants of this line were crossed with wild-type plants, the abortion rate was 26.2% (*n* = 504, Fig. 1A). These results indicate that *cip111* mutation also affects ovule development. This was confirmed by observations that 22.9% of the *cip111/+; mScarlet-CIP111m/-* female gametophytes (*n* = 35) had arrested at developmental stage FG2 or FG3[23] (Fig. 1C). We also performed Alexander staining[24] to investigate the viability of pollen from *cip111/+; mScarlet-CIP111m/-* and wild-type plants. The results revealed that 23.1% of the pollen grains (*n* = 359, Supplementary Fig. 2C, F) from *cip111/+; mScarlet-CIP111m/-* plants stained green, an indicator of lethality, while only 0.8% of pollen grains (*n* = 236) from wild-type plants stained green (Supplementary Fig. 2A, F). Of the *cdc48D/+* pollen grains 4.8% stained green (*n* = 146, Supplementary Fig. 2B, F), and the in vitro pollen germination rate was lower (60%, *n* = 519, Supplementary Fig. 2E, G) than that of the wild type (88%, *n* = 532, Supplementary Fig. 2D, G). Combined these results demonstrate that both *CIP111* and *CDC48D* are required for male and female gametophyte development and plant reproduction.

### *OAP1, CDC48D* and *CIP111* are predominantly expressed in meristematic tissues

YFP fusion proteins driven by the corresponding native promoter were used to investigate the expression patterns of *OAP1, CDC48D* and *CIP111* in Arabidopsis tissues. The constructs *pOAP1:OAP1-YFP*, *pCDC48D:YFP-CDC48D* and *pCIP111:YFP-CIP111m* were transformed into *oap1, cdc48D* and *cip111* mutants, respectively. The phenotypes observed in the mutants could be rescued by expression of the corresponding transgenes, which established that they are functional. Prominent fluorescence signals from OAP1-YFP, YFP-CDC48D, and YFP-CIP111 were observed in the root meristem and lateral root primordium of stably transformed Arabidopsis seedlings, while weak signals were detected from other tissues (Fig. 2A and Supplementary Fig. 7C). Thus, it can be concluded that *OAP1, CDC48D* and *CIP111*−similar to OPNR[19]−also are strongly expressed in meristem tissue with active cell proliferation.

### Inducible CRISPR/Cas9 mutants demonstrate that *OAP1, OAP2, CDC48D* and *CIP111* have essential roles in vegetative development

The OPNR-YFP[19], OAP1-YFP, YFP-CDC48D and YFP-CIP111 signals observed in roots indicate that these proteins have roles that go beyond gametophyte and seed development. To investigate this, we first needed to overcome the obstacle of how gametophyte and embryo lethal mutants can be subjected to functional analysis. For this reason, we generated inducible CRISPR/Cas9 mutants of *OPNR, OAP1,* *OAP2, CDC48D* and *CIP111*. The constructs were created based on the published β-estradiol inducible XVE system[25,26] and plant specific CRISPR/Cas9 system[27,28], with certain modifications. The Arabidopsis *RIBOSOMAL PROTEIN S5A* (*pRPS5A*) promoter, which is highly expressed in meristematic cells[29], was used to drive the expression of XVE[30] induced Cas9 (*pRPS5A:XVE:Cas9*). The inducible CRISPR/Cas9 construct targeting *OAP2* was transformed into wild type Col-0 plants, while the constructs targeting *OPNR, OAP1, CDC48D* and *CIP111* were introduced into *opnr/opnr; pOPNR:OPNR-mScarlet, oap1/oap1; pOAP1:OAP1-mScarlet, cdc48D/cdc48D; pCDC48D:mScarlet-CDC48D* and *cip111/cip111; pCIP111:mScarlet-CIP111m* complemented lines, respectively. Complementation of the mutated native genes with mScarlet fusion transgenes enabled reductions in the mScarlet signal to serve as an in vivo marker for the CRISPR/Cas9-facilitated mutation of the target gene.

The inducible CRISPR/Cas9 (*icr*) mutants were grown on solid growth media for five days and then transferred to media with 5 μM β-estradiol to induce the expression of the Cas9. The mScarlet signals were prominent in the root meristems at the time of transfer (Fig. 2B) but declined or disappeared three days after Cas9 induction (Fig. 2B). At the same time, meristem size[31] and root growth rate clearly decreased in all of the *icr* lines relative to the wild-type counterparts (Fig. 2B, C). Taken together, these results showed that *OPNR, OAP1, OAP2, CDC48D* and *CIP111* are important for both meristem maintenance and vegetative development in *Arabidopsis*.

### OAP1, CDC48D, and CIP111 colocalize with OPNR on the nuclear envelope and mitochondria

We previously showed that OPNR exhibits an unusual subcellular localization, namely, that it is present at both the nuclear envelope and mitochondria[19]. In order to determine whether OAP1, CDC48D, and CIP111 show similar subcellular localizations we crossed the various fluorescent protein fusion lines. The subsequent analysis of these lines showed that OAP1-YFP colocalizes with both OPNR-mScarlet (Fig. 3A) and mScarlet-CIP111m (Fig. 3B), with OPNR-YFP and mScarlet-CIP111m also showing similar localization (Fig. 3C). Crosses with the mitochondrial marker line mt-CFP[32] and mit-GFP[33] revealed that OAP1-YFP and mScarlet-CIP111m also colocalize with mt-CFP and mit-GFP, respectively; this was also previously observed for OPNR-YFP (Supplementary Fig. 3A, B)[19]. The nuclear envelope localization of OAP1 and CIP111 was also confirmed by the colocalization of nuclear envelope markers NUP54-CFP and SUN2-CFP[19] with OAP1-YFP and mScarlet-CIP111, respectively (Supplementary Fig. 3D, E). These results indicate that OAP1 and CIP111 colocalize with OPNR on the nuclear envelope and mitochondria, as well as imply that these proteins can interact.

We also crossed mScarlet-CDC48D with OPNR-YFP. The mScarlet and YFP signals demonstrated overlap in the cytosol (Fig. 3D), but the OPNR-YFP signal at the nuclear envelope was not as clear as what had been observed for lines with only OPNR-YFP (Fig. 3C). The offspring of the cross between mScarlet-CDC48D and OPNR-YFP included lines which expressed only OPNR-YFP; in these lines, the OPNR-YFP signal at the nuclear envelope was restored (Supplementary Fig. 3F). This observation indicates that the co-expression of mScarlet-CDC48D affects the localization of OPNR-YFP. To further investigate this dynamic, we expressed YFP-CDC48D under a β-estradiol inducible promoter in the OPNR-mScarlet and OAP1-mScarlet background. Before induction, the OPNR-mScarlet and OAP1-mScarlet signals at the nuclear envelope and cytosol were clear (Fig. 3E and Supplementary 3G). However, after the induction of YFP-CDC48D expression, the nuclear envelope signals of OPNR-mScarlet and OAP1-mScarlet reduced markedly relative to the cytoplasmic signal (Fig. 3F and Supplementary 3H). The subcellular localization of YFP-CIP111 was also affected by the co-expression of mScarlet-CDC48D (Supplementary Fig. 3I). This noticeable effect of YFP-CDC48D expression on the

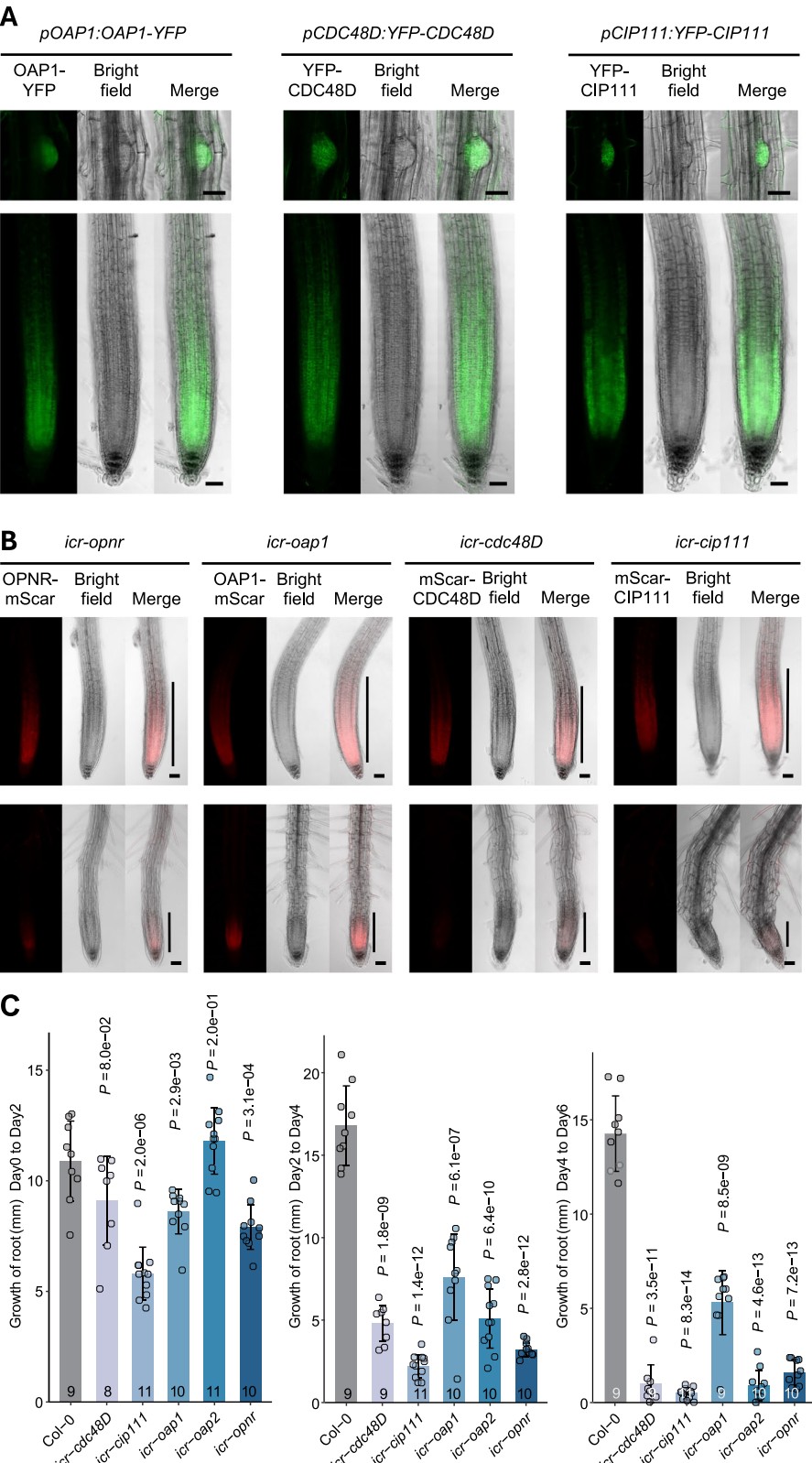

**Fig. 2 | *OAP1, CDC48D* and *CIP111* are predominantly expressed in the meristem and are important for meristem maintenance. A** Expression of *pOAP1::OAP1-YFP*, *pCDC48D:YFP-CDC48D* and *pCIP111:YFP-CIP111* in root tips and lateral root primordia. **B** The inducible CRISPR/Cas9 lines of *opnr* (*icr-opnr*), *icr-oap1, icr-cdc48D, icr-cip111* before (upper panels) and after (lower panels) Cas9 induction. The vertical lines in the Figs denote the root meristems. **C** Root growth of Col-0, *icr-opnr,* *icr-oap1, icr-oap2, icr-cdc48D* and *icr-cip111* lines. The growth was measured every two days at 2, 4 and 6 days after Cas9 induction. Data represent means ± SD. The *P* values indicate the significance of differences between mutants and Col-0 control based on a two-tailed Student's *t*-test. The number of replicates are shown at the bottom of each column. Scale bars: 50 μm.

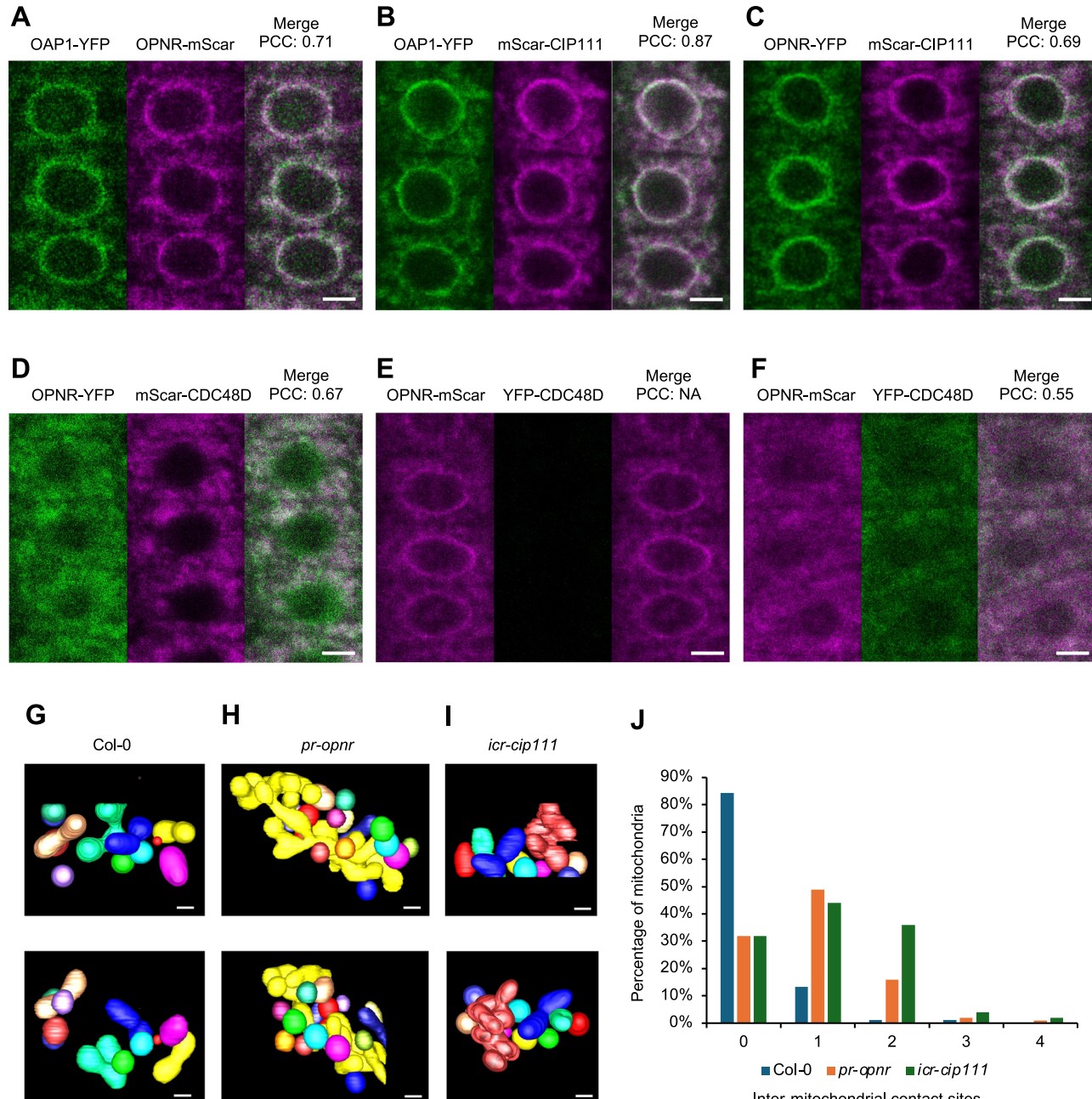

**Fig. 3 | Subcellular localization of OAP1, CDC48D and CIP111. A–D** Confocal laser scanning microscope (CLSM) images of root tip cells from seedlings stably expressing *pOPNR:OPNR-mScarlet* and *pOAP1:OAP1-YFP* (A), *pOAP1:OAP1-YFP* and *pCIP111:mScarlet-CIP111m* (B), *pOPNR:OPNR-YFP* and *pCIP111:mScarlet-CIP111m* (C), and *pOPNR:OPNR-YFP* and *pCDC48D:mScarlet-CDC48D* (D). The degree of colocalization was determined by the Pearson correlation coefficient (PCC). **E, F** CLSM images of root tip cells from seedlings harboring *pRPS5A:XVE:YFP-CDC48D* and *pOPNR:OPNR-mScarlet* before (**E**) and after (**F**) induction of the CRISPR/Cas9 system. The same settings of YFP and mScarlet channels were used for (**E**) and (**F**). There are almost no YFP-CDC48D signals before inducing in (**E**). PCC was also shown for (**E**). NA, not applicable. **G–I** Three-dimensional reconstructions of mitochondria, based on focused ion beam scanning electron microscopy (FIB-SEM) of Col-0 (**G**), *pr-opnr* (**H**) and *icr-cip111* (**I**) root meristem cells. Sequential SEM images were captured at 30 nm intervals, with each mitochondria contour labeled and reconstructed into a 3D shape. Mitochondria were classified based on their matrix connectivity, with each distinct mitochondria marked in different colors. **J** The number of inter-mitochondrial contact sites between neighboring mitochondria. *n* = 83 for Col-0, *n* = 100 for both *pr-opnr* and *icr-cip111*. Scale bars: (**A–F**): 5 μm; (**G–I**): 0.5 μm.

subcellular distribution of OPNR, OAP1 and CIP111 provides additional evidence that these proteins can interact with each other.

To further investigate the cytoplasmic localization of CDC48D, we crossed mScarlet-CDC48D with the mitochondrial marker mit-GFP. Subsequent imaging revealed that the mScarlet-CDC48D signal was partly colocalized with the mit-GFP signals (Supplementary Fig. 3C). To investigate if OPNR localize to the membrane or matrix of mitochondria, we co-expressed OPNR-YFP with the mitochondrial membrane marker prohibitin3-mCherry (PHB3-mCherry)[19] and the mitochondrial matrix marker mt-CFP[32] (Supplementary Fig. 3J). Mutation of *PHB3* is known to reduce plant growth and cause enlarged mitochondria, while the overexpression of *PHB3* also affects plant growth[34]. Here we found that the lines expressing PHB3-mCherry also contain some enlarged mitochondria, and that OPNR-YFP clearly colocalizes with PHB3-mCherry at the periphery and is not found in the matrix of these enlarged mitochondria (Supplementary Fig. 3J). Similarly, OAP1-YFP

and YFP-CDC48D also showed mitochondrial periphery localization on the enlarged mitochondria in the PHB3-mCherry expressing lines (Supplementary Fig. 3K, L). These results indicate that OPNR, OAP1 and CDC48D are localized on the periphery of mitochondria.

To confirm the mitochondria localization of CDC48D and other OPNR interactors, we performed GFP-trap based co-IP and TurboID based proximity labelling analysis on mitochondria isolated from callus expressing TurboID-YFP-CDC48D or TurboID-YFP. Mitochondria were isolated from the biotin treated callus prior to protein extraction. The results from both co-IP and TurboID analysis showed that OPNR, OAP1, OAP2, CDC48D and CIP111 were all enriched in TurboID-YFP-CDC48D compared to the TurboID-YFP samples. Mitochondrial outer membrane proteins like TOM20-1, TOM20-2, TOM20-3, TOM20-4 and MIRO-RELATED GTP-ASE 1 (MIRO1) were also enriched in TurboID-YFP-CDC48D samples (Supplementary Fig. 4D, E, Supplementary Data 1, 2). Thus, taken together these results provide further support for the localization of OPNR and its interactors on mitochondria.

We previously observed smaller mitochondria in the meristematic root tip cells of Arabidopsis *opnr* mutants, which suggests that OPNR has a role in mitochondrial morphogenesis and function[19]. To investigate the size and shape of mitochondria in more detail, we used focused ion beam scanning electron microscopy (FIB-SEM) to observe the root meristem cells of the original *pr-opnr* partial rescued line[19] and the *icr-cip111* mutants. The FIB-SEM 3D reconstruction of mitochondria revealed a connected network of mitochondria in both *pr-opnr* and *icr-cip111* mutants, while such a network was not observed in the control lines (Fig. 3G-I). Furthermore, inter-mitochondrial contact sites are also increased in the *pr-opnr* and *icr-cip111* lines (Fig. 3J and Supplementary Fig. 4A-C). The subcellular localization and FIB-SEM results – when considered together – point to a functional role of the OPNR and the associated proteins on both nuclear envelope and mitochondria.

### OPNR, OAP1, OAP2, CDC48D, and CIP111 form an AAA+ ATPase complex

We performed co-IP experiments on the crude protein extracts isolated from free YFP, OPNR-YFP, OAP1-YFP, YFP-CDC48D, and YFP-CIP111 complemented Arabidopsis lines of corresponding mutants. The mass spectrometry analysis showed an enrichment of OPNR, OAP1, OAP2, CDC48D and CIP111 in the OPNR-YFP, OAP1-YFP, YFP-CDC48D and YFP-CIP111 complemented lines in contrast to the free YFP expressing control line (Supplementary Fig. 5A–D and Supplementary Data 3). The TurboID proximity labelling and GFP-Trap co-IP experiments with CDC48D also showed that these five proteins are highly enriched in the mitochondria fraction compared with the control (Supplementary Fig. 4D, E and Supplementary Data 1-5). These results suggested that OPNR, OAP1, OAP2, CDC48D, and CIP111 are part of the same protein complex in vivo.

To determine whether OPNR, OAP1 and OAP2 can directly interact with each other, we simultaneously expressed these three recombinant proteins in *E. coli* and discovered that all three proteins co-purified with the N-terminal 6xHis-tagged OAP1 at a 1:1:1 molar ratio of OPNR:OAP1:OAP2 (Fig. 4A). After removal of the 6xHis-tag, the molecular weight of the OPNR-OAP1-OAP2 complex was determined by Size Exclusion Chromatography (SEC). The results revealed that the size of the complex is ~80 kDa, which is close to the molecular weight that can be calculated from the individual molecular weights of OPNR (23 kDa), OAP1 (34 kDa), and OAP2 (28 kDa) trimer with 1:1:1 ratio (85 kDa in total, Supplementary Fig. 6A). These results established that OPNR, OAP1, and OAP2 form a trimer with 1:1:1 ratio.

The protein structure prediction tool AlphaFold was used to model the three-dimensional structure of the OPNR-OAP1-OAP2 trimer. The results showed that, despite a lack of amino acid sequence homology, OPNR, OAP1, and OAP2 share similar structures (Supplementary Fig. 6B), and the tool also provided high confidence scores (ipTM = 0.79, pTM = 0.8, Fig. 4B) for these proteins interacting as a trimer. Furthermore, the modelling tool was also used to assess the interaction between the OPNR-OAP1-OAP2 trimer and CIP111. The results showed that the N-terminus of CIP111 could interact with the OPNR-OAP1-OAP2 trimer based on the low predicted aligned error (PAE) score between the N-terminus of CIP111 and the trimer (Fig. 4C-F)[35]. The CIP111 N-terminus alone was also used to predict the interaction with the OPNR-OAP1-OAP2 trimer. The results showed high confidence scores (ipTM = 0.72, pTM = 0.75, Fig. 4E), which provides further support for the interaction.

How CDC48D and CIP111 interact and their stoichiometry in the complex is not known, but published structural studies of similar proteins provide some clues. CDC48D and CIP111 show high amino acid sequence similarity with the large evolutionary ancient AAA+ ATPase family known to form homo- or heterohexamers[36–38]. Both CDC48D and CIP111 contain characteristic AAA+ ATPases domains D1 and D2 involved in ATP binding and hydrolysis[21,22]. Previous studies of mammalian CDC48D/p97 indicated that the nucleotide binding in the D1 domain is important for the complex formation[39,40]. Thus, in order not to exceed AlphaFold's amino acid residue limit only the D1 and D2 domain sequences were used to model CDC48D and CIP111 interaction and stoichiometry. The results suggested that the proteins could form homo- and heterohexamers with similar ipTM and pTM scores (Supplementary Fig. 6C-F). We also used the recently developed deep learning-based 3D structural modeling tool DeepMSA2-based protein folding (DMFold), which has shown significant improvements relative to AlphaFold in predicting protein complex structures[41]. Due to the tool's amino acid sequence limit, only homo- or heterodimers formed by the D1 and D2 domains of CDC48D and CIP111 were modelled. In these predictions the CDC48D-CIP111 heterodimer showed higher pTM score (0.51) than CDC48D-CDC48D (pTM = 0.33) and CIP111-CIP111(pTM = 0.38) homomers (Supplementary Fig. 6G-I). To further investigate the putative interaction between CDC48D and CIP111 in vivo, we used a low background bimolecular fluorescence complementation (BiFC) assay[42] in *Nicotiana benthamiana* leaves. Clear fluorescence signals were observed in the leaf epidermal cells co-expressing nYFP-CDC48D and cYFP-CIP111, but not in the controls (Supplementary Fig. 6J). Thus, both the modelling and BiFC results support a CIP111 and CDC48D heterohexamer composition of three CDC48D-CIP111 heterodimers.

Finally, the OPNR-OAP1-OAP2 trimer was modelled together with the CDC48D-CIP111 heterodimer, with the prediction demonstrating that the trimer interacts with the N-terminus of CIP111, and CDC48D interacts with the D1 and D2 domains of CIP111 (Fig. 4F). Next, we attempted to model the entire complex formed by the OPNR-OAP1-OAP2 trimer and CDC48D-CIP111 hexamer. Due to the amino acid sequence length limit of the AlphaFold, only the N-terminus and D1 domain of CIP111 and CDC48D were included in the prediction. The resulting model illustrates that the three N-terminus of CIP111 in the CDC48D-CIP111 heterohexamer occupies the three bends of the OPNR-OAP1-OAP2 trimer, and that the pore of the OPNR-OAP1-OAP2 trimer is aligned with the central tunnel of the hexamer (Supplementary Fig. 6K). Taken together, these results revealed that OPNR, OAP1 and OAP2 form a trimer which can interact with the N-terminal domains of the CDC48D-CIP111 heterohexamer to form a heteromeric AAA+ ATPase protein complex, called the OPNR complex hereafter.

### The OPNR complex is required for RPL24C extraction from the pre-60S ribosomes

To elucidate the function of the OPNR complex, the predicted 3D structures of OPNR, OAP1 and OAP2 were input into the protein structure comparison database Dali (http://ekhidna2.biocenter.helsinki.fi/dali/). This revealed that the human C1ORF109 and CDK2-interacting protein (CINP) are structurally similar to OPNR and OAP1, respectively (Supplementary Fig. 7A and Supplementary Data 4, 5). Interestingly, two of the human AAA+ ATPases, namely SPATA5 and

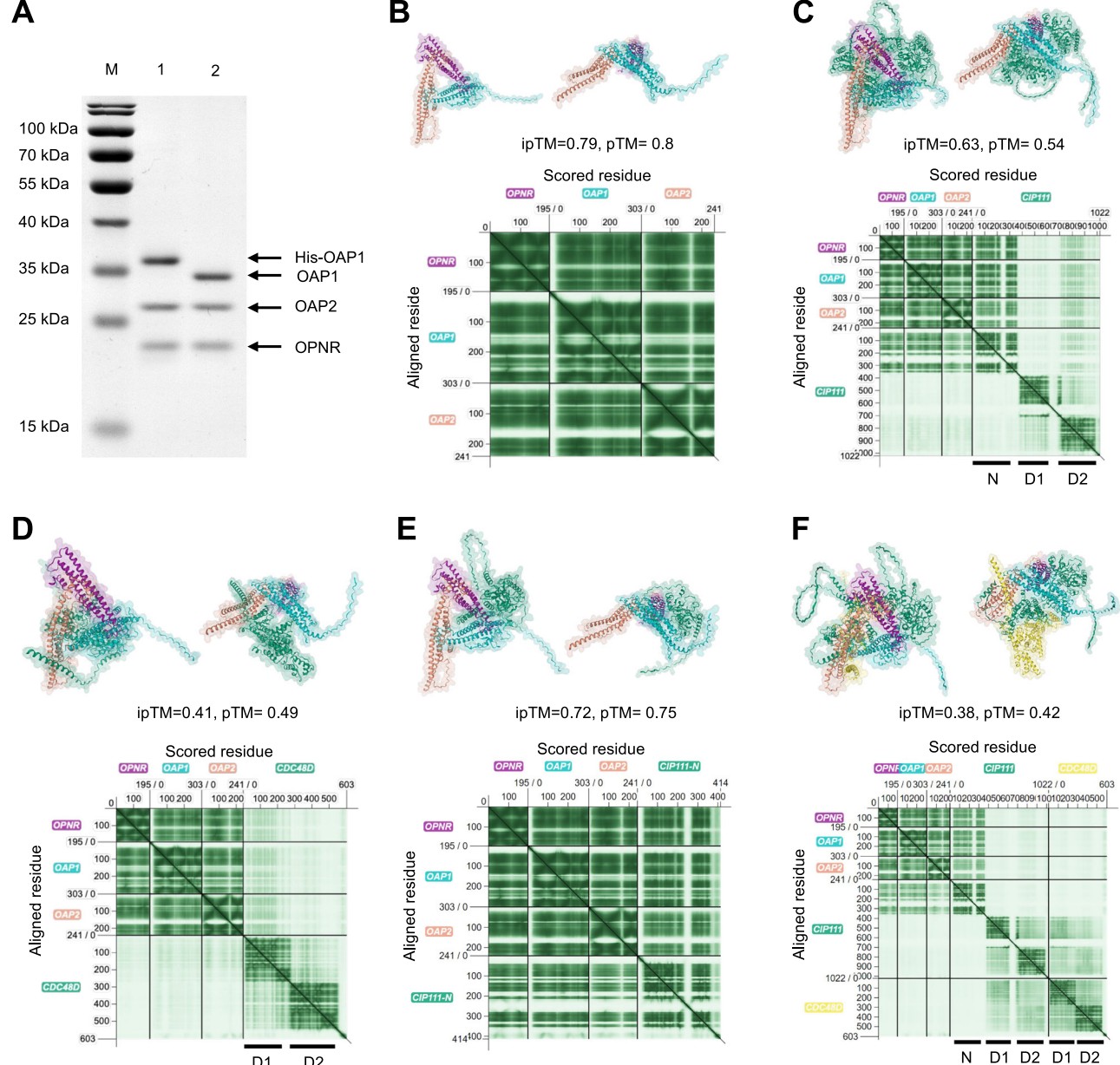

**Fig. 4 | Structural modeling of the OPNR complex formed by OPNR, OAP1, OAP2, CDC48D, and CIP111. A** The purified recombinant proteins His-OAP1, OAP2 and OPNR before (lane 1) and after (lane 2) His-tag removal. The molar ratio measured for OAP1: OAP2: OPNR was 1.2: 1: 1.1. **B**–**F**: AlphaFold predicted structures of the OPNR-OAP1-OAP2 trimer (**B**), the interaction between OPNR-OAP1-OAP2 trimer and CIP111 (**C**), the interaction between OPNR-OAP1-OAP2 trimer and CDC48D (**D**), the interaction between OPNR-OAP1-OAP2 trimer and the N-terminus of CIP111 (**E**), and the interaction between OPNR-OAP1-OAP2 trimer, CIP111 and CDC48D (**F**). In

each Fig., the three-dimensional structure is shown from two different angles. The predicted template modelling (pTM) score and the interface predicted template modelling (ipTM) score is shown for each prediction. The predicted aligned error (PAE) plots provided for each predicted structure demonstrate the high confidence (dark green) and low confidence (pale green) regions for the predicted structure. The colour coding indicates the protein identity as shown in the PAE plots. The N-terminus domain of CIP111 (N) and the two ATPase domains (D1 and D2) of CIP111 and CDC48D were labelled.

SPATA5L1, were recently shown to interact with C1ORF109 and CINP[43]. The D1 and D2 ATPase domains of SPATA5 and SPATA5L1 are similar to those of CIP111 and CDC48D, respectively. Experimental evidence from human cell cultures suggests that C1ORF109, CINP, SPATA5, and SPATA5L1 form a complex that is required for 60S ribosome assembly[38,43]. For instance, the ribosome assembly factor RSL24D1, which is normally recycled from the cytosol to the nucleus, was retained in the cytosol in the loss-of-function mutants of C1ORF109, CINP, SPATA5 and SPATA5L1[43]. Thus, it is believed that human C1ORF109, CINP, SPATA5 and SPATA5L1 form a complex that is required for RSL24D1 removal from the pre-60S ribosome in the cytosol.

Based on the human cell culture results, we hypothesised that the Arabidopsis homolog of RSL24D1 could be a substrate for the OPNR complex. Amino acid sequence comparison showed that the Arabidopsis RPL24C is the closest homolog of RSL24D1 (60% identity and 78% similarity over 143 amino acid residues, Supplementary Fig. 7B). Moreover, Arabidopsis RPL24C and human RSL24D1 are also the closest homologs of the *Saccharomyces cerevisiae* 60S ribosome assembly factor Rlp24, although both have shorter C-terminal sequences (Supplementary Fig. 7B). The possibility that Arabidopsis RPL24C interacts with the heterotrimer consisting of OPNR, OAP1 and OAP2 was first investigated using AlphaFold. The prediction stated that RPL24C may interact with the

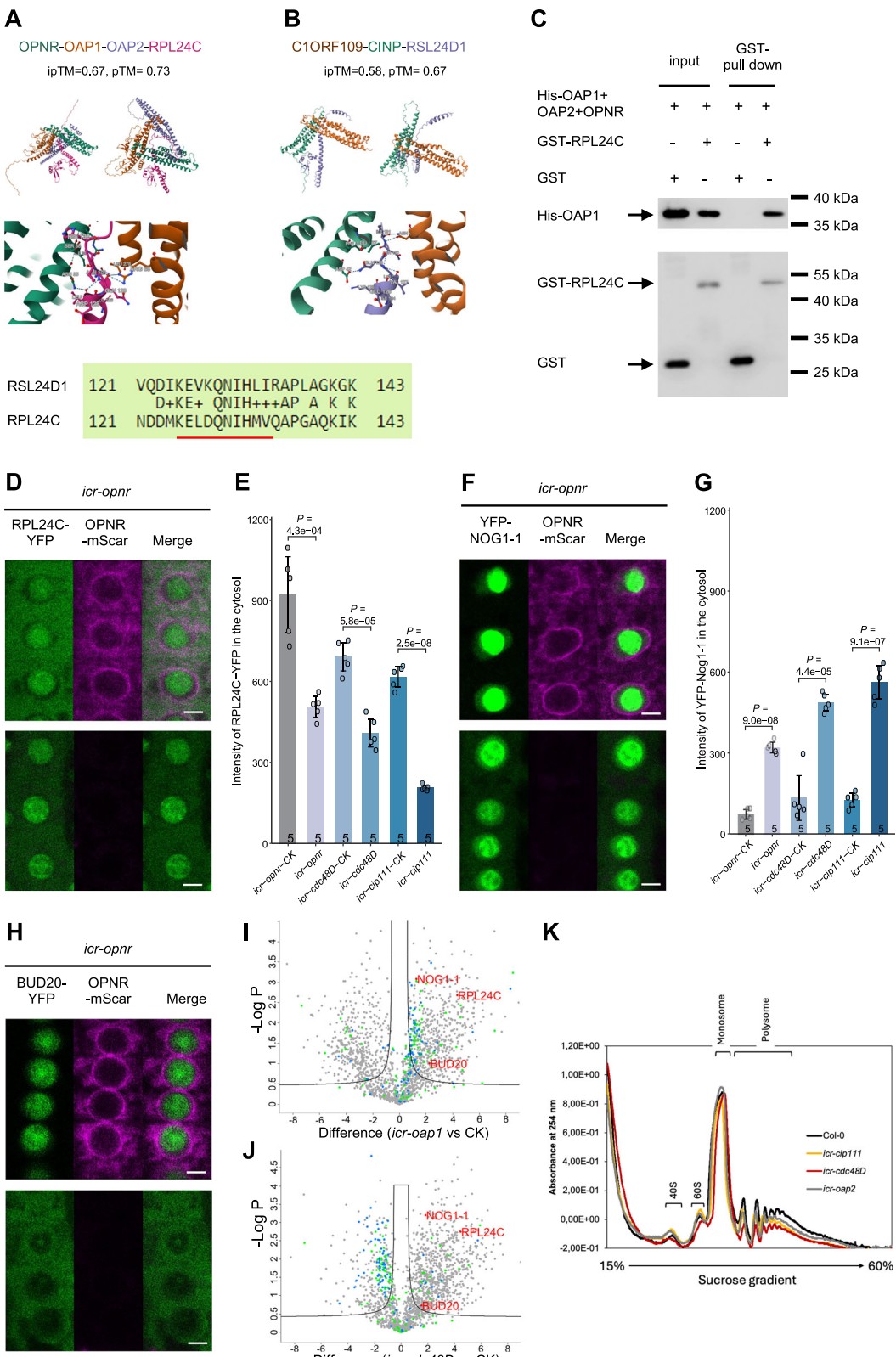

OPNR-OAP1-OAP2 trimer based on high confidence scores (ipTM = 0.67, pTM = 0.73) (Fig. 5A). In the model, the C-terminus of RPL24C is placed into the central pore of the OPNR-OAP1-OAP2 trimer (Fig. 5A). Similarly, the RSL24D1 was also predicted to interact with C1ORF109 and CINP (ipTM = 0.67, pTM = 0.73, Fig. 5B). Interestingly, the contact sites at which Arabidopsis RPL24C and human RSL24D1 were predicted to interact with the OPNR-OAP1-OAP2

trimer and C1ORF109-CINP dimer (Fig. 5A, B), respectively, involve similar amino acids. This finding suggests an evolutionarily conserved mechanism. The interaction between RPL24C and the OPNR-OAP1-OAP2 trimer was also experimentally verified by an in vitro pull-down assay between recombinant His-OAP1-OPNR-OAP2 and GST-RPL24C proteins (Fig. 5C). Thus, RPL24C is a likely substrate for the OPNR complex.

**Fig. 5 | RPL24C serves as the substrate for the OPNR complex. A, B:** AlphaFold predicted structures and the interaction between the Arabidopsis OPNR-OAP1-OAP2 trimer and RPL24C (**A**), and the human C1ORF109-CINP dimer and RSL24D1 (**B**). The contact sites and conserved sequences are also shown. **C** Recombinant protein pull-down assay results for GST-RPL24C and the His-OAP1-OPNR-OAP2 trimer. Free GST proteins served as the control. 5% input used for the His-OAP1-OPNR-OAP2 trimer and 100% input used for the GST-RPL24C and GST. **D** CSLM images show the localization of RPL24C-YFP before (upper panel) and after (lower panel) inducible CRISPR/Cas9-facilitated mutation of *OPNR* (*icr-opnr*). The same settings of YFP and mScarlet channels were used for upper and lower panels. The OPNR-mScarlet signals almost disappeared after 3 days of inducing in the lower panel. **E, G** Bar charts show measurements of the RPL24C-YFP (**E**) and YFP-NOG1-1 (**G**) signals in the cytosol before and after CRISPR/Cas9 induced mutation of *OPNR*, *CDC48D* and *CIP111*. Data represent mean of five measured cells ± SD. The *P* values

were generated from a two-tailed Student's *t*-tests. **F, H** Localization of YFP-NOG1-1 (**F**) and BUD20-YFP (**H**) before (upper panel) and after (lower panel) CRISPR/Cas9-induced mutation of *OPNR* (*icr-opnr*), respectively. **I, J** Proteomic analyses of the 60S fractions from mutant and wild-type plants. Volcano plot results for *icr-oap1* vs CK (**I**) and *icr-cdc48D* vs CK (**J**) with -log10 *P*-value on the y-axis and log2 intensity differences on the x-axis. *P*-values were calculated using two-tailed Student's *t*-test, moderated by Benjamini–Hochberg's method. FDR = 0.05, s0 = 0.5. The intensity of each protein was normalized to the total signal of each sample. Each dot represents a protein with the green dots representing proteins that are part of large ribosome subunits and blue dots representing small ribosome subunits. RPL24C, NOG1-1 and BUD20 are shown in red. The full lists are shown in Supplementary Data 6. **K** Sucrose density gradient centrifugation obtained ribosome profiles for wild-type, *icr-oap2*, *icr-cip111* and *icr-cdc48D* calli after CRISPR/Cas9-induced mutation. Scar bars: 5 μm.

In *Saccharomyces cerevisiae*, Rlp24 is removed from the pre-60S ribosomes by the AAA+ ATPase Drg1, upon which it is recycled back to nucleus as the pre-60S ribosome is transported to the cytosol via a nuclear pore[6,10]. Mutation of Drg1 results in cytosolic retention of Rlp24[10,11]. Furthermore, mutations of the human C1ORF109, CINP, SPATA5 and SPATA5L1 also caused the cytosolic retention of RSL24D1[43]. These results prompted us to investigate whether the Arabidopsis RPL24C is also retained in the cytosol following OPNR complex defects. For this experiment, a C-terminal YFP fusion of RPL24C (*pRPL24C:RPL24C-YFP*) under the control of the native promoter was introduced into the inducible CRISPR/Cas9 mutants of *OPNR*, *CDC48D* and *CIP111*. Prior to the Cas9 induction, the RPL24C-YFP signal was prominent in root meristem cells (Supplementary Fig. 7C), with an expression pattern that was similar to what was observed for the components of the OPNR complex (Fig. 2A). The RPL24C-YFP signal, when analyzed in the meristematic cells, showed similar intensity in the nucleolus and the cytosol (Fig. 5D, Supplementary 7D, E). Three days after Cas9 induction, the RPL24C-YFP signal in the root meristem cells was significantly reduced in the cytosol relative to what was observed for the control (Fig. 5D, E, Supplementary 7D, E). This is the opposite of what could be expected if the RPL24C release from the pre-60S ribosome was inhibited. However, we noted that the assay was not well-suited for assessing the cytosolic retention of RPL24C-YFP in the cytosol because of the prominent cytosolic RPL24C-YFP signal that was observed already prior to Cas9 induction. To overcome this problem, we searched for alternative markers of RPL24C release.

In *Saccharomyces cerevisiae*, the assembly factors Nog1 and Bud20 also associate with the pre-60S ribosomes and interact with Rlp24; moreover, the release of both Nog1 and Bud20 depends on the removal of Rlp24 by Drg1[10,11]. Consequently, mutations in Drg1 will result in Nog1 and Bud20 continuing to be associated to the pre-60S ribosomes in the cytosol, and thus results in the cytosolic retention of these proteins[10,44]. Thus, we investigated the localization of the Arabidopsis homologs of Nog1 and Bud20[1] in the inducible OPNR complex mutants. Arabidopsis NOG1-1 shows the highest similarity to *Saccharomyces cerevisiae* Nog1[45], and the Arabidopsis protein encoded by *AT2G36930* shows the highest similarity to *Saccharomyces cerevisiae* Bud20 (35% identities and 50% positives of 110 out of total 199 amino acids); this protein was thus named BUD20. Both YFP-NOG1-1 and BUD20-YFP showed clear cytosolic retention after CRISPR/Cas9 induced mutations in *OPNR* and the associated interacting proteins (Fig. 5F-H, Supplementary Fig. 7F, G). We also examined the localization of RPL24C-YFP, YFP-NOG1-1 and BUD20-YFP in response to leptomycin B, which is an inhibitor of nuclear export[46]. Leptomycin B treatment caused the retention of RPL24C-YFP, YFP-NOG1-1 and BUD20-YFP in the nucleus (Supplementary Fig. 7H-J), establishing that RPL24C, YFP-NOG1-1 and BUD20-YFP are exported from the nucleus to the cytosol. These results corroborated that the release of NOG1-1 and BUD20 and thus likely also RPL24C, depends on the OPNR complex.

To verify the retention of the ribosome biogenesis factors on the 60S ribosomes, we used sucrose density gradient centrifugation to isolate the 60S ribosome fraction from the cytosol of wild type callus and *icr-oap1*, *icr-cdc48D* inducible mutant callus 6 days after CRISPR/Cas9 induction. Proteomics mass spectrometry analysis of the 60S fraction showed that RPL24C, NOG1-1 and BUD20 were all enriched in the *icr-oap1* and *icr-cdc48D* lines relative to the control line (Fig. 5I, J, Supplementary Fig. 7K, and Supplementary Data 6). These data imply a defect in how the release of these placeholders from the pre-60S ribosomes in the cytosol is governed.

Defective 60S ribosome assembly would be expected to have a negative effect on ribosome biogenesis. To investigate this, we isolated cytosolic and nuclear ribosomes from wild-type and *icr-oap2*, *icr-cip111*, and *icr-cdc48D* callus four days after CRISPR/Cas9 induction (Fig. 5K). The sucrose density gradient centrifugation profiles showed that the polysome level decreased in all three mutants relative to the wild type, while the amount of free 40S ribosomes increased in the *icr-oap2* and *icr-cip111* mutants, and the 60S ribosomes in the *icr-cip111* mutant (Fig. 5K). Since OPNR, OAP1 and OAP2 function as a heterotrimer and the corresponding mutants show almost identical phenotypes in ovule and seed development, the *icr-oap2* based conclusions are likely to be representative of the heterotrimer defects. These observations corroborate defective ribosome assembly in the mutants, and - when combined with the 60S fraction proteomics results – confirm that the loss-of-function of the OPNR complex results in the cytosolic retention of RPL24C, NOG1-1, and BUD20 on the pre-60S ribosomes. Thus, defective ribosome biogenesis is a likely explanation for the lethal phenotypes observed in the mutant plants.

## Discussion

We identified and characterized a heteromeric AAA+ ATPase protein complex named here the OPNR complex, which is required for 60S ribosome assembly in plants. A previously identified, evolutionarily conserved protein (OPNR)[19], and two other structurally similar proteins (OAP1 and OAP2) form a heterotrimer which interacts with the N-terminal domains of the AAA+ ATPase heterohexamer formed by CIP111 and CDC48D. Evidence for the proteins that are involved in this specific complex is based on evidence which demonstrates similar mutant phenotypes, similar gene expression patterns, similar subcellular localizations, verified protein interactions, and structural models. Alterations in the subcellular signals of 60S ribosome assembly factors in the mutants provided strong evidence that the OPNR complex is required for ribosome biogenesis. The cytosolic retention of the ribosome assembly factors RPL24C, NOG1-1 and BUD20 on pre-60S ribosomes in response to defects of the OPNR complex provided further support of its role in ribosome biogenesis. Combined the results support a model wherein the OPNR-OAP1-OAP2 trimer acts as an adapter required for the CDC48D-CIP111 AAA+ ATPase binding to RPL24C and its subsequent release from the pre-60S ribosome (Fig. 6).

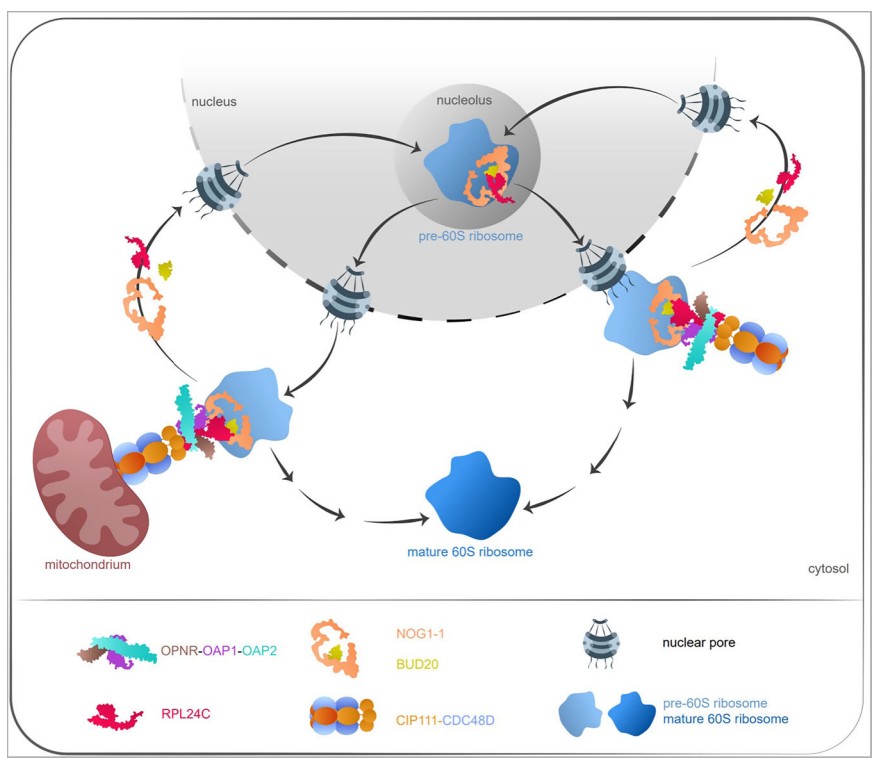

**Fig. 6 | Schema showing the function of the OPNR complex in 60S ribosome assembly.**

The functional role of OPNR was initially revealed by co-immunoprecipitation and mutant analysis. For instance, two of the investigated mutants, *oap1* and *oap2*, showed an early seed abortion phenotype previously observed in *opnr* (Fig. 1A); furthermore, as was the case with *opnr*, the nucleoli of mature ovules and zygotes in the *oap1* and *oap2* lines were noticeably larger than what was observed for the wild type (Fig. 1B, C). The arrested zygotes of *oap1* and *oap2* demonstrated the same apical vacuole phenotype as was observed for *opnr* (Fig. 1C)[19]. The almost identical phenotypes of *oap1*, *oap2* and *opnr* are consistent with the fact that OPNR, OAP1 and OAP2 form a heterotrimer with 1: 1: 1 ratio (Fig. 4A-C). The AAA+ ATPase mutants *cip111* and *cdc48D* also showed female and male gametophyte defects. The phenotype witnessed in these mutant lines was more severe than what was observed in the *opnr*, *oap1* and *oap2* mutant lines (Fig. 1 and Supplementary Fig. 2); this indicates that CIP111 and CDC48D may have several cofactors in addition to the OPNR-OAP1-OAP2 trimer, and consequently, other substrates/targets in addition to the RPL24C. Prior research supports this theory as CDC48A, a close homolog of CIP111 and CDC48D in *Arabidopsis* has been shown to have several cofactors and substrates[47–49]. It is also possible that the CIP111 and CDC48D ATPase alone can extract RPL24C from the 60S pre-ribosome, but that this extraction occurs more efficiently in conjunction with the OPNR-OAP1-OAP2 trimer. In contrast, Drg1, the *Saccharomyces cerevisiae* homolog of CIP111, is able to directly interact with Rpl24, the homolog of RPL24C in *Saccharomyces cerevisiae*[10]. The mRNAs and/or the proteins encoded by *CIP111* and *CDC48D* may be also less stable than the *OPNR*, *OAP1* and *OAP2* counterparts, a dynamic which could result in early gametophyte defects[50,51]. In line with this hypothesis, the inducible CRISPR/Cas9 mutants of *CIP111* and *CDC48D* showed an earlier decrease in mScarlet-CIP111m and mScarlet-CDC48D fluorescence, respectively, along with more severe root growth defects when compared to the *OPNR*, *OAP1* and *OAP2* mutants (Fig. 2B, C).

When compared to male gametophyte development, more defects in female gametophyte and embryo development were observed among the *opnr*, *oap1*, *oap2* and *cdc48D* mutants (Fig. 1 and

Supplementary 2). Similar observations were noted in other Arabidopsis ribosome biogenesis mutants called SLOW WALKER1 (SWR1), SLOW WALKER2 (SWR2), SLOW WALKER3 (SWR3) and NUCLEOLAR FACTOR1 (NOF1). SWR1 is a nucleolar protein containing a WD40-domain that is involved in pre-rRNA processing and 18S pre-rRNA biogenesis in Arabidopsis[52]. SWR2 is the homolog of *Saccharomyces cerevisiae* NUCLEOLAR COMPLEX ASSOCIATED PROTEIN1 (NOC1), with both involved in the export of pre-ribosomes from the nucleus to the cytosol[53]. SWR3 is a DEAD-box RNA helicase that is required for 18S rRNA biogenesis[54], while NOF1 is an evolutionarily conserved protein needed for rRNA biogenesis[55]. Mutations in these genes/proteins lead to embryo sac development arrest at different stages with a less dramatic effect on male gametophyte development. This observation may be connected to the fact that the female gametophyte–when compared to the male gametophyte–undergoes more mitotic cell divisions, has a larger cell size, and demonstrates cellularization processes that result in ribosome dilution and higher demands on ribosome biogenesis[56]. It is important to state that ribosome biogenesis is also important for male gametophyte development and meristem maintenance[56–58]. AtNOB1 and AtENP1 are components of the 40S pre-ribosome and function as co-factors for 40S ribosome biogenesis in Arabidopsis. AtNOB1 is an endonuclease cleaving pre-rRNA. Mutants of *AtNOB1* and *AtENP1* show delayed pollen development and reduced transmission of the mutation to the next generation through both female and male gametophytes[56]. The AAA-ATPase Midasin 1 (MDN1) is essential for 60S ribosome biogenesis and plays a crucial role in root meristem maintenance. Mutations of MDN1 lead to defects in ribosome assembly and affect root development and meristem maintenance[58]. NOTCHLESS (NLE) is involved in 60S ribosome biogenesis and interacts with MDN1. *nle* mutants also show defects in root meristem maintenance and embryo development[57]. In this study we also observed pollen defects in *cdc48D* and *cip111* mutants (Supplementary Fig. 2) and the decreased root meristem size in the inducible CRISPR/Cas9 *opnr*, *oap1*, *oap2*, *cdc48D* and *cip111* mutants (Fig. 2B, C). Thus, these

phenotypes are consistent with the role of ribosome biogenesis in gametophyte development and meristem maintenance.

In our experiments, *OPNR, OAP1, CIP111* and *CDC48* showed the highest expression in undifferentiated meristematic cells, which is consistent with functions in ribosome biogenesis and meristem maintenance (Fig. 2A)[19]. Genes encoding for additional ribosome bio-genesis factors also demonstrate heightened expression in the mer-istems and play a role in meristem maintenance. Nuclear/Nucleolar GTPase 2 (Nug2) is a GTPase with functions in the 27S pre-ribosomal RNA (rRNA) processing required for 60S ribosome biogenesis. Strong *AtNug2* expression has been observed in meristematic regions like root tips and shoot apexes, with RNAi mutants of *AtNug2* are more sensitive to the translational inhibitor cycloheximide[59]. Midasin/Ribosome export/assembly (MDN1/Rea1) is a large, dynein-related AAA-type ATPase that is required for the removal of Nug2 prior to the nuclear export of pre-60S ribosomes[60]. Furthermore, in Arabidopsis *MDN1* demonstrates strong expression in the meristem, more specifically, in the shoot apex and root tip[61]. Accordingly, a weak allele of *mdn1-1* showed reduced root meristem, short roots, dwarf shoots and reduced seed set[61], while the null mutant *mdn1-2* is embryonic lethal[58]. The large subunit GTPase 1 (LSG1), which is encoded by two genes *LSG1-1* and *LSG1-2*, plays a role in the late maturation of the 60S ribosome. *LSG1-2* is highly expressed in newly emerged leaves, root tips, and lateral root primordia[62], while *lsg1-2* mutants have shorter roots, less lateral roots, and overall smaller size. Moreover, *lsg1-1, lsg1-2* null mutants are lethal, which implies that LSG1 plays a critical role in the 60S ribosome maturation associated with meristem activity[63]. Thus, *OPNR, OAP1, OAP2, CDC48D* and *CIP111* expression in the meristems, along with the meristem defects observed in the CRISPR/Cas9-induced mutants, are consistent with functions in ribosomal biogenesis and meristem maintenance in Arabidopsis.

The modeling and protein interaction results confirmed that OPNR, OAP1, OAP2, CDC48D and CIP111 form a complex. As displayed in Fig. 4D, OPNR, OAP1, and OAP2 form a heterotrimer reminiscent of a three-finger claw with a pore in the center. According to the structural modeling, the OPNR-OAP1-OAP2 trimer interacts with the N-terminal domain of CIP111 (Fig. 4E), with the N-terminus of CIP111 occupying the bend formed by OPNR and OAP1 (Fig. 4E). CDC48D and CIP111 belong to a large, evolutionarily conserved AAA+ ATPase family which is known to form homo- and heterohexamers[36-38]. A heterohexameric complex is in line with similar phenotypes of the *cdc48D* and *cip111* mutants, and the structural model prediction of a heterohexamer made of three CIP111 and CDC48D heterodimers (Supplementary Fig. 5). According to a recently published structure resolved via cryo-EM, the human SPATA5 and SPATA5L1 AAA+ ATPase homolog also forms a heterohexamer[38]. Thus, the current results–when considered together with the latest empirical evidence–suggest that heteromeric AAA+ ATPases share similar structural features across complex mul-ticellular eukaryotes.

The modeling further indicated that the OPNR-OAP1-OAP2 trimer mediates the interaction between CDC48D-CIP111 and the ribosome assembly factor RPL24C in Arabidopsis. In the modeling runs, the C-terminal tail of RPL24C was placed into the pore of the OPNR-OAP1-OAP2 trimer and the whole protein under the trimer claw (Fig. 5A). Subsequent in vitro experiments confirmed the direct interaction between recombinant proteins. Interestingly, structural modelling of the human homologs suggests that RSL24D1, the human homolog of RPL24C, also interacts with the dimer formed by C1ORF109 and CINP, which are the homologs of the Arabidopsis OPNR and OAP1, respectively (Fig. 5B). Notably, the amino acids present in the predicted contact site are conserved across human and Arabidopsis proteins (Fig. 5A, B). These results suggest that a similar, AAA+ ATPase and co-factor catalyzed mechanism exists in humans and Arabidopsis plants for the release of placeholders RSL24D1 and RPL24C from cytoplasmic pre-60S ribosomes. In *Saccharomyces cerevisiae*, this function is carried out by Drg1, which is a homolog of CIP111 and CDC48D. Drg1 forms a homohexamer in which the N-termini of Drg1 interacts with the C-terminal end of Rlp24 to release it from the pre-60S ribosome[6]. The Rlp24 C-terminal domain also stimulates Drg1 ATP hydrolysis, with the extraction of Rlp24 from the pre-60S ribosomes catalyzed by the D2 domain and the release of Rlp24 from Drg1 dependent on the D1 domain[10,64]. In contrast to *Saccharomyces cerevisiae*, the critical roles of the C1ORF109 and CINP dimer and OPNR-OAP1-OAP2 trimer in human and Arabidopsis AAA+ ATPase complexes, respectively, suggest addi-tional complexity, and possibly regulatory layers, in the 60S ribosome biogenesis that occurs across complex multicellular eukaryotes. One aspect of this complexity may be linked to the subcellular distribution of the AAA+ ATPase complex in Arabidopsis.

Experiments with fusion proteins in which OPNR, OAP1 and CIP111 were tagged with a fluorescent protein revealed that these proteins primarily localize to the nuclear envelope and mitochondria (Fig. 3A-C). The nuclear envelope localization of the OPENER complex is con-sistent with its function in RPL24C extraction from the pre-60S ribo-some when it emerges through the nuclear pore. The *Saccharomyces cerevisiae* Drg1 interacts with nucleoporins of the cytosolic part of the nuclear pore complex, and the release of Rlp24 from the pre-60S ribosome requires the interaction of Drg1 with the nucleoporins. Drg1 is hence also likely localized to the nuclear envelope, although the overexpressed Drg1 localized to cytoplasm and did not support this interpretation[11]. This may have, however, been a consequence of the high Drg1 expression level in this study. Overexpression may also have influenced the results of C1ORF109 and SPATA5 localization in human and mouse cells, respectively[65-67] since none of these studies reported signal enrichment on the nuclear envelope.

We found that–in Arabidopsis–RPL24C is localized to the nucleolus, nucleus, and cytosol (Fig. 5D and Supplementary Fig. 6D, E and H). The cytosolic localization of RPL24C is also supported by proteomic studies of cytosolic ribosomes in Arabidopsis[68,69]. Similarly, the *Saccharomyces cerevisiae* homolog Rlp24 also showed cytosolic localization under standard growth conditions[11,70]. The human homo-log RSL24D1, when investigated under standard conditions and in wild-type cells, showed predominantly nucleolus localization, although a weak cytosolic signal is also evident in the published images[43]. The cytosolic YFP-RPL24C signal observed in the present study could ori-ginate from free RPL24C and/or RPL24C that is associated with pre-60S ribosomes, or even monosomes and polysomes. Based on studies in *Saccharomyces cerevisiae* and human cell lines, Rlp24 and RSL24D1 are mainly associated with pre-60S/60S ribosomes fractions[43,70]. However, in the *c1orf109*^KO and *SPATA5*^KO mutant cells, RSL24D1 was not only found to be associated with 60S ribosomes, but also with 80S monosomes and even with polysomes[43].

It is clear that NOG1-1 and BUD20 show different subcellular dis-tribution after mutation of the OPNR complex components (Fig. 5F, H and Supplementary Fig. 7F, G). In *Saccharomyces cerevisiae*, Nog1 and Bud20 are both cofactors for 60S ribosome assembly, but Nog1 joins the pre-60S ribosome early in the nucleolus, while Bud20 joins pre-60S later in the nucleoplasm[71]. The Nog1 mutant in *Saccharomyces cerevi-siae* is lethal[15], while Bud20 mutant exhibits a slow-growth phenotype[17]. Thus, it is possible that NOG1-1 and BUD20 in Arabidopsis play dif-ferent roles in pre-60S ribosome assembly and behave differently when the function of OPNR complex is disrupted, although in this research we found that both NOG1-1 and BUD20 are prominently localized to nucleolus in the wild-type root tip cells (Fig. 5F, H, Sup-plementary Fig. 7I, J). Furthermore, it is also possible that NOG1-1 and BUD20 have distinct synthesis and turnover rates which could lead to different localization in the *icr-opnr* and *icr-cdc48D* mutants. Homologs of the ribosome assembly factors RPL24C, NOG1 and BUD20 in *Sac-charomyces cerevisiae* and humans have been shown to shuttle between the nucleus and cytosol[10,11,15-17,43,64,72]. Our results point to similar dynamism in plants: NOG1-1 and BUD20 were localized to the

nucleus in the control but showed cytosolic retention in the OPNR complex mutants (Fig. 5F–H). This indicated that these factors are recycled from cytosol to nucleus in the WT control. The LMB treatment experiments also showed that RPL24C, NOG1-1 and BUD20 are exported from the nucleus in the wild-type control (Supplementary Fig. 7H–J). In summary, the results in this study together with the *Saccharomyces cerevisiae* and human results corroborate the shuttle of the ribosome assembly factors between nucleus and cytosol.

The mitochondria localization of the OPNR complex is surprising, but literature evidence supports the presence of ribosome on mitochondria and thus also the possibility of ribosome biogenesis. Conforming with this idea, the mouse AAA+ ATPase SPATA5, also known as spermatogenesis-associated factor (SPAF), contains a mitochondrial matrix targeting sequence and the protein localizes to mitochondria[67]. It has been suggested that mouse SPATA5/SPAF is involved in mitochondrial morphogenesis during spermatogenesis[67], with knock-down in neurons affecting mitochondrial morphology and dynamics[66]. In humans, mutations in SPATA5 have also been connected to mitochondrial disorders[44,73–75]. In cultured human cells, overexpressed human SPATA5 mainly localized to cytosol and did not overlap with a mitochondria marker[66]. The human C1ORF109 was shown to localize to nucleus and cytoplasm using immunolabeling[65]. In our previous study we found that when overexpressed OPNR is also localized to both nucleus and cytoplasm, and a fixed sample of native promoter expressed OPNR-YFP was also dispersed in nucleus and cytoplasm (Supplementary Fig. 3M, N). Thus, it remains possible that the native SPATA5 and C1ORF109 also localize to nuclear envelope and mitochondria. Furthermore, it is possible that the subcellular distribution of SPATA5 and C1ORF109 may require native tissue context, which was missing in the human and mouse cell culture experiments.

In our experiments, we used native promoters to drive the expression of fluorescent protein fused OPNR, OAP1, CDC48D and CIP111 which complemented the phenotype of the corresponding mutants. This allowed us to investigate the in vivo localization of these proteins in the native tissue context. Our results clearly showed that the cytoplasmic signal of OPNR, OAP1, CDC48D and CIP111 predominantly localize to mitochondria. Furthermore, we also observed changes in the mitochondria morphogenesis in the *pr-opnr* and *icr-cip111* mutants supporting an important role for mitochondrial function for the OPNR complex (Fig. 3G–J). Supplementary Fig. 3J–L show that OPNR-YFP, OAP1-YFP and YFP-CDC48D predominantly localize to the periphery of mitochondria. The proximity labeling results showed that the mitochondria outer membrane TOM20 proteins are in the vicinity of the CDC48D and other components of the OPNR complex (Supplementary Fig. 4E) further supporting the OPNR complex localization to the outer surface of mitochondria. Thus, the OPNR complex catalyzed pre-60S ribosome maturation may also occur on the surface of mitochondria. We hypothesize that this may guarantee local translation capacity within a cell. There is convincing evidence that several nuclear-encoded mitochondrial proteins destined for mitochondria are translated on the mitochondrial surface. For example, mRNAs for the nuclear encoded voltage-dependent ion channel (VDAC) and other mRNAs with a cis-element in the 3' untranslated region (3' UTR) localize to mitochondria in Arabidopsis[76,77]. In *Saccharomyces cerevisiae*, the RNA-binding protein Puf3p, which is involved in mRNA transport, preferentially binds mRNAs that encode nuclear-produced mitochondrial proteins implying mRNA targeting to and translation on mitochondria[78]. Studies in *Saccharomyces cerevisiae* have also shown that mRNAs related to ribosome-nascent chain (NAC) complexes associated mRNAs and the mitochondrial receptor for cytosolic ribosomes are also co-translationally imported into mitochondria[79,80]. The mammalian RNA binding protein Clu1/CluA homologue (CLUH) binds mRNAs of nuclear encoding mitochondrial proteins and loss of CLUH reduced the level of these mRNAs and caused mitochondria defects[81]. The Arabidopsis homolog of CLUH, FRIENDLY MITOCHONDRIA

(FMT), was also suggested to bind RNAs and cytosolic ribosomes at the surface of mitochondria[82]. Direct evidence of the co-translational import of mitochondrial proteins came from the electron cryo-tomography observation that cytosolic ribosomes are present on the surface of isolated mitochondria[83]. TOM20 proteins are mitochondrial import receptor subunits that were suggested to mediate local translation coupled with import of some nuclear encoded mitochondrial proteins[84]. Thus, ribosomes and translation on the mitochondrial surface have been documented in several studies. Our results suggest that the subcellular localization of the final stages of 60S ribosome biogenesis is an important prerequisite for this process. This direct association with mitochondria may also be important for ensuring sufficient ATP availability for the ribosome biogenesis and the subsequent protein translation. In line with this hypothesis local translation in hippocampal neurons has been suggested to be powered by mitochondria nearby[85].

The fluorescent protein fusions of OPNR, OAP1 and CIP111 all show similar localization on the nuclear envelope and mitochondria in the presence of the native CDC48D. It follows that the native CDC48D proteins should also localize to nuclear envelope and mitochondria since OPNR, OAP1, OAP2, CDC48D and CIP111 can form a complex. This conclusion was also supported by the fact that the N-terminal fusion of YFP or mScarlet to CDC48D affected the distribution of OPNR, OAP1 and CIP111 (Fig. 3D–F and Supplementary Fig. 3F–I). In these lines the nuclear envelope localization was not clear anymore, while the mitochondria association was still prominent. Thus, it seems that the N-terminal fusion of YFP or mScarlet altered the distribution of CDC48D and associated proteins for some reason. Interestingly the mScarlet-CDC48D fusion protein could complement the ovule arrest phenotype of *cdc48D* as shown in Fig. 1A, which raises the possibility that the mitochondria associated 60S ribosome assembly catalyzed by the OPNR complex can be sufficient.

The role of OPNR complex in ribosome biogenesis is evident by the changes in 40S and 60S fractions the higher free ribosome to mature translating ribosome (mono- and polysome) ratio following OPNR complex mutation, although the changes were not uniform across the mutants (Fig. 5K). The mutants of different OPNR complex components showed also other phenotypic differences. Ovule and seed development was impaired in *opnr*, *oap1* and *oap2* mutants at a similar stage (Fig. 1A, C), while the *cdc48D* and *cip111* showed more severe phenotypes in both ovule and pollen development. In addition, the pollen and ovule development in *cip111* was arrested earlier compared to *cdc48D* as shown in Fig. 1B and Supplementary Fig. 2A. These differences suggest that there may be additional cofactors and substrates of CDC48D and CIP111. Furthermore, the samples used for ribosome profile analysis were derived from inducible CRISPR/Cas9 mutants, and the efficiency of inducible CRISPR/Cas9 knockout varies between different components of the OPNR complex as shown in Fig. 2C.

The role of OPNR complex in ribosome biogenesis is also supported by the enlarged nucleoli in the mutants (Fig. 1B, C)[19]. Similar enlarged nucleoli were also oberseved in the Arabidopsis mutants *thallo* (*thal*)[86], *yaozhe* (*yao*)[87] and *rna helicase10* (*rh10*)[88] defective in ribosome biogenesis. Thus, taken together it can be concluded that the vital role of the OPNR complex is linked to its function in ribosome biogenesis. The mechanism of the subcellular targeting of the OPNR complex and its role in maintaining translation capacity warrants further study, in particular since the current evidence suggests that it may be a common feature of multicellular eukaryotes. In comparison to yeast our results reveal increased pre-ribosome processing complexity in plants, and although the OPNR complex has similarities to its human counterpart, the complex has not been observed on both nuclear envelope and mitochondria in other organisms. Especially the mitochondria localization of the OPNR complex and the defective mitochondria morphology in the mutants of the OPNR complex point to previously

unrecognized subcellular complexity of ribosome biogenesis in plants and mitochondria association as a feature to ensure sufficient translational capacity. Our work emphasizes the importance of studying ribosome biogenesis in different organisms. The overall process appears conserved in eukaryotes, but our results reveal differences that warrant further investigation. In a wider context, ribosomes are important determinants of crop yield and performance, and our work provides a new opening and tools to study this important topic.

# Methods

## Plant material and growth conditions

The *Arabidopsis thaliana* ecotype Columbia-0 (Col-0) was selected as a wild-type control. Seeds of T-DNA insertion lines, namely, *opnr-1* (SALK_148287) and *oap1* (SALK_042474), were ordered from the Nottingham Arabidopsis Stock Centre (Nottingham, UK). The plants were grown in soil at 22 °C and 65% relative humidity. The photoperiod is 16 h of light and 8 h of dark, with a light intensity of 150 μmol m$^{-2}$ s$^{-1}$. All the primers used in this study are listed in Supplementary Table 1.

The *oap2*, *cip111*, and *cdc48D* mutants were generated by gene editing of the *OAP2*, *CIP111* and *CDC48D* genes in the Col-0 background using the CRISPR/Cas9 system and three pairs of guide RNA (gRNA) targets: OAP2-sg1/OAP2-sg2, CDC48D-sg1/CDC48D-sg2, and CIP111-sg1/ CIP111-sg2, respectively (Supplementary Table 1). The gRNAs sequences were cloned into the pHEE401E vector as previously described[27,89]. Next, the constructs were introduced into Col-0 by Agrobacterium-mediated transformation[90]. The CRISPR/Cas9 system-induced mutations in the *OAP2*, *CDC48D* and *CIP111* genes were identified via PCR using the primer pairs oap2-Fw/oap2-Rv, cdc48D-Fw/cdc48D-Rv, and cip111-Fw/cip111-Rv, respectively (Supplementary Fig. 1, Supplementary Table 1). The CRISPR/Cas9 transgenes were then eliminated from the *oap2* and *cdc48D* mutants by backcrossing with Col-0 plants. In case of *cip111* mutants, N-terminal mScarlet was fused to gRNA targeting site mutated *CIP111* (*pCIP111:mScarlet-CIP111m*) and transformed into *cip111* CRISPR mutants carrying CRISPR/Cas9 transgenes. Next, lines with *cip111* mutations and *pCIP111:mScarlet-CIP111m* transgenes, but without CRISPR/Cas9 transgenes were selected from the progeny for further experiments.

For inducible CRISPR/Cas9 system-facilitated gene editing, the EC1.2-EC1.1 fusion promoter (EC1fp) in the pHEE401E[89] vector was replaced with the pRPS5A-XVE inducible promoter[29,30] to generated the pH5AXVE vector. When creating the *OPNR*, *OAP1*, *OAP2*, *CDC48D*, and *CIP111* mutants, the guide RNA pairs OPNR-sg1/OPNR-sg2, OAP1-sg1/OAP1-sg2, OAP2-sg1/OAP2-sg2, CDC48D-sg1/CDC48D-sg2 and CIP111-sg3/CIP111-sg4, respectively, were used (Supplementary Table 1). Next, the constructs OPNR-pH5AXVE, OAP1-pH5AXVE, OAP2-pH5AXVE, CDC48D-pH5AXVE, and CIP111-pH5AXVE were transformed into *opnr/opnr*; *pOPNR:OPNR-mScarlet*, *oap1/oap1*; *pOAP1:OAP1-mScarlet*, Col-0, *cdc48D/cdc48D*; *pCDC48D:mScarlet-CDC48D* and *cip111/cip111*; *pCIP111:mScarlet-CIP111m* plants, respectively. The transgenic lines were grown vertically on half Murashige and Skoog (MS) agar plates for five days and then transferred to half MS agar plates complemented with 5 μM β-estradiol.

For the nuclear export inhibition experiments, 5 days old seedings were treated with 10 μg/mL leptomycin B (Beyotime) in half MS medium for 5 h before observation.

## Phenotypic analysis

Wild-type and mutant siliques were dissected with tweezers and needles under a Leica MZ 9.5 stereomicroscope (Leica, Wetzlar, Germany). Ovules or seeds were excised and mounted with Hoyer's solution[20]. After several hours of clearing, samples were observed under a Zeiss Axioplan 2 microscope equipped with a Zeiss AxioCam HRc camera (Zeiss, Oberkochen, Germany). Differential interference contrast optics were used. Images were captured and processed by ZEN2011 software (Zeiss).

## Complementation of the mutants

For complementation experiments involving *oap1*, the 1053-bp promoter sequence upstream of the *OAP1* start codon was amplified. The 1053-bp *OAP1* promoter sequence and the *OAP2* coding sequence were then fused with *mScarlet* or *mNeonGreen* into the pGREEN 0029 or pGREEN 0179 vectors[91] by Gibson cloning (New England Biolabs, Ipswich, MA) to create the *pOAP1:OAP1-mScarlet* and *pOAP1:OAP1-mNeonGreen* constructs. These constructs were then introduced into *oap1* mutant plants, after which plants homozygous for both T-DNA insertions and transgenic genes (*oap1/oap1*; *OAP1-mScarlet* and *oap1/oap1*; *OAP1-mNeonGreen*) were obtained in the T3 generation. The siliques were checked under a Leica MZ 9.5 stereomicroscope (Leica). For complementation experiments involving *oap2*, the 2238-bp *OAP2* promoter sequence and *OAP2* coding sequence were amplified and fused into the pGREEN 0029 vector by Gibson cloning to create the *pOAP2:OAP2* construct. The construct was transformed into the *oap2* mutant, after which plants homozygous for both *oap2* mutation and the introduced transgene (*oap2/oap2*; *pOAP2:OAP2*) were obtained. For complementation experiments involving *cdc48D*, the 1914-bp *CDC48D* promoter, *mScarlet*, and *CDC48D* coding sequences were amplified and fused into the pGREEN 0029 vector by Gibson cloning to create the *pCDC48D:mScarlet-CDC48D* construct. The construct was transformed into the *cdc48D* mutant, after which plants homozygous for both the *cdc48D* mutation and the introduced transgene (*cdc48D 2/ cdc48D*; *mScarlet-CDC48D*) were obtained. For complementation experiments involving *cip111*, the 1900-bp *CIP111* promoter, *mScarlet*, and *CIP111* coding sequences with mutated sgRNA target sites were amplified and fused into the pGREEN 0029 vector by Gibson cloning to create the *pCIP111:mScarlet-CIP111m* construct. The construct was transformed into the *cip111* mutant that had been created via CRISPR/Cas9 gene editing, after which plants homozygous for both the *cip111* mutation and the introduced transgene (*cip111/ cip111*; *mScarlet-CIP111m*) were obtained. For *pOPNR:OPNR-mScarlet*, the *OPNR* promoter and *OPNR* coding sequence were amplified from the *pOPNR:OPNR-YFP* construct[19] and fused with *mScarlet* into the pGREEN 0029 vectors using Gibson cloning. For *pOAP1:OAP1-YFP*, *pCDC48D:mScarlet-CDC48D* and *pCIP111:mScarlet-CIP111m* constructs, the GreenGate[92] system was used. The promoters and coding sequences were cloned into pGGA and pGGC modules, respectively. The YFP coding sequence was cloned into pGGB and pGGD modules for N-terminal and C-terminal fusions, respectively. The resulting constructs were transformed into the respective mutant plants.

## Protein localization analysis

To determine the subcellular localization of the proteins, seedlings were grown vertically on half MS plates. The roots were mounted in liquid half MS medium and observed using either a Zeiss LSM780 or a Zeiss LSM880 confocal laser scanning microscope. The default settings for eCFP, eGFP, eYFP, mCherry, and chlorophyll B were used when detecting CFP, GFP, YFP, mScarlet, and chlorophyll signals, respectively. The images were captured and processed using the ZEN 2.3 SP1 black edition software (Zeiss). *pUBQ10:NUP54-CFP* was construct by the GreenGate[92] system. *NUP54* CDS and CFP coding sequences were amplified and cloned into the pGGC and pGGD modules, respectively. The degree of colocalization was determined by the Pearson correlation coefficient (PCC) determined by Coloc2 in ImageJ/Fiji software.

## Recombinant protein expression and pull-down assay

A pACYCDuet-1 construct with 6xHis-OAP1, OAP2 and OPNR or pET-GST construct with GST-RPL24C were expressed in the *E.coli* strain BL21(DE3) cultured in LB broth. To induce the expression 0.4 mM isopropyl 1-thio-β-d-galactopyranoside (IPTG) was introduced when the OD600 reached 0.6-0.7, after which the culture was allowed to grow over night at 18 °C. Bacteria were harvested by centrifugation and

lysed using sonication. The His-OAP1-OAP2-OPNR proteins were purified on Ni-NTA resin (Thermo Fisher Scientific), followed by elution with 200 mM imidazole or cleavage of the His-tag by using HRV 3C-protease and re-purified through the Ni-NTA resin (Thermo Fisher Scientific) to remove the 6xHis-tag and 3C-protease. The GST-RPL24C proteins were purified on Glutathione Sepharose High Performance beads (Cytiva) and eluted with Elution buffer (50 mM Tris-HCl, 10 mM reduced glutathione, pH 8.0). Finally, the proteins were purified on a Superdex200 16/600 column (Cytiva). For pull-down assay, the same molecular amount of GST (sigma) and GST-RPL24C were incubated with His-OAP1-OAP2-OPNR 30 mins at 4 °C, next Glutathione Sepharose High Performance beads were added to the mixture and incubated for 1 h at 4 °C. The beads were washed with washing buffer (50 mM Tris-HCl, 150 mM NaCl, pH 8.0) for three times. The beads were boiled and the released proteins analysed by Western-blotting.

## Bimolecular fluorescence complementation assays
The *CDC48D* and *CIP111* full-length cDNA were fused with nYFP210 and cYFP210[42] coding sequences drivien by the UBQ10 promoter with the GreenGate[92] cloning system to generate nYFP-CDC48D and cYFP-CIP111 fusion constructs, respectively. The appropriate constructs were introduced into *Agrobacterium tumefaciens* strain GV3101 together with the pSOUP vector[92] and then transiently expressed in *Nicotiana benthamiana* leaves by infiltration as described[93]. The transformed leaves were observed under a Zeiss LSM980 confocal laser scanning microscope.

## Mitochondria isolation
Crude mitochondria isolation was performed as described previously[94]. In short, one gram of callus were ground in a mortar on ice with 5 ml of extraction buffer (10 mm EDTA, 60 mm TES, 10 mm KH2PO4, 0.3 m sucrose, 1 mm glycine, 1% polyvinylpyrrolidone, 25 mm Na4P2O7, 1% BSA, pH adjusted with KOH to 8.0 and 50 mm sodium ascorbate, plus 20 mm cysteine added prior to grinding, pH readjusted to 8.0 with KOH) and 0.5 g of quartz sand. The ground sample and 5 ml extraction buffer used to rinse the mortar were transferred filtered through a 20 μm nylon mesh. The filtered homogenate was then centrifuged 2500 g 5 min at 4 °C. The supernatant was transferred into a new tube and centrifuged 15 000 g 15 min at 4 °C. The pellet was washed with 5 mL wash buffer (2 mm EDTA, 10 mm TES, 10 mm KH2PO4, 0.3 m sucrose and pH adjusted with KOH to 7.5) and centrifuged at 15 000 g for 15 min at 4 °C. The mitochondria enriched pellet was used for further experiments.

## TurboID proximity labelling
TurboID proximity labelling was performed according to following a published protocol[95]. Briefly, TurboID-YFP-CDC48D and TurboID-YFP calli were grown in liquid half MS medium. 50 μM biotin were added to the medium for 3 h for the labeling. The biotin containing medium was removed and samples were rinsed with ice cold water for 5 min for three times to stop the labeling reaction and remove excess biotin. The samples were then used for mitochondria isolation as described above.

Proteins were extracted from the mitochondria enriched fraction with extraction buffer (50 mM Tris pH 7.5, 150 mM NaCl, 0.1% SDS, 1% Triton-X-100, 0.5% Na-deoxycholate, 1 mM EGTA, 1 mM DTT, 1x complete, 1 mM PMSF). The suspension was centrifuged 15,000 g 15 min at 4 °C. Then the supernatant was filtrated with Amicon Ultra-0.5 Centrifugal Filter 3 kDa (Millipore) 3 times to remove excess free biotin. Protein concentration was measured by Bradford (BioRad protein assay). 16 mg total protein was incubated with 100 μL Dynabeads MyOne Streptavidin C1 (Thermo Fisher Scientific) in 5 ml protein LoBind tube (Eppendorf). Complete protease inhibitor (Thermo Fisher Scientific) was also added, and the samples were incubated at 4 °C overnight on a rotor. Then the beads were separated on magnetic rack and washed with the following solutions: 2x with cold extraction buffer, 1x with cold 1 M KCl and 1x with 2 M Urea in 10 mM Tris pH 8 at room temperature and 2x extraction buffer without complete protease inhibitor. Finally, the beads were used for on-beads trypsin digestion and followed by proteomics mass spectrometry.

## GFP-Trap immunoprecipitation and protein digestion
Total proteins from the pPRS5A:YFP, pOPNR:OPNR-YFP, pOAP1:OAP1-YFP, pCIP111:OAP1-YFP and pCDC48D:CDC48D-YFP calli were extracted on ice with ice cold extraction buffer (50 mM Tris, pH 7.5, 150 mM NaCl, 1 mM EDTA, 0.5% Triton X-100, 0.5% IGEPAL CA-630, 10% [v/v] glycerol, 1 × Complete protease inhibitor cocktail (Roche, Basel, Switzerland). For TurboID-YFP-CDC48D and TurboID-YFP samples, extraction buffer was used to resuspend the mitochondria enriched pellet to extract the proteins. Extracted proteins were incubated with GFP-Trap-MA beads (ChromoTek, Planegg, Germany) on a rotator for 30 min at 4 °C. The beads were then washed five times with extraction buffer. The proteins that were bound to the beads were eluted with elution buffer (0.1 M glycine pH2.5, 1% Triton X-100) for 30 sec with pipetting. Next, 5% v/v 1 M Tris base was added immediately to bring the pH to ~7.5, after which the eluted proteins were digested using a modified SP3 protocol[96,97]. Briefly, SpeedBeads A and B magnetic carboxylate modified particles (beads A hydrophilic, GE45152105050250; beads B hydrophobic, GE65152105050250, Sigma Aldrich, St. Louis, MO) were mixed at a 1:1 (v/v) ratio and washed four times using LC-MS grade water. Then, the beads were added to each sample, contained in the binding buffer (50% ethanol and 2.5% formic acid as final concentration), and incubated at 500 rpm for 15 min at room temperature. The samples were then transferred into a filter plate (0.22 μm, MSGVN2210, Sigma Aldrich). The unbound fraction was discarded by centrifugation at 1000 g for 5 min. Beads retained on the filter were washed four times with 70% ethanol. Then digestion buffer (100 mM HEPES pH7.5, 5 mM chloroacetamide, 1.2 mM TCEP) with trypsin was added to each sample (0.2 μg trypsin) on the plate. Digestions were performed at room temperature with shaking at 500 rpm overnight. Flowthrough was collected in collection plate with centrifugation at 1000 g for 5 min. Beads were washed and eluted again with 10 μl 2% DMSO and the flowthrough was pooled with the previous fraction. Peptides in the solution were desalted on an Oasis HLB plate (cat. No.186001828BA, Waters Corporation, Milford, MA) according to the manufacturer's instructions and then dried by speed vac.

## Mass spectrometry and data analysis
Dried peptides were dissolved in 20 μL 0.1% formic acid in water. A 5 μL aliquot of peptides from each sample was injected into the mass spectrometer using the Vanquish Neo system (Thermo Fisher Scientific, Waltham, MA). The trapping column PEPMAP NEO C18 (5 μm particle size, 300 μm*5 mm, Thermo Fisher Scientific) was used, while the analytical column was a nanoEaseTM M/Z HSS C18 T3 (100 Å, 1.8 μm particle size, 75 μm*250 mm; Waters). The total length of the experimental run was 90 min; separation and elution started with 98% mobile phase A (0.1% formic acid in water) and 2% mobile phase B (80% acetonitrile and 0.1% formic acid), which rose to 8% B over 4 min, and to 27% B over 60 min, and then to 40% B over 13 min. After that, the proportion of mobile phase B was increased to 80% in 0.1 min and held for 4 min, with the proportion of mobile phase B dropping to 2% B in 0.5 min. This cycle was followed by column equilibration.

Data acquisition on the Exploris 480 system (Thermo Fisher Scientific) was carried out by using a data dependent method. Mass range coverage of 375 – 1500 were acquired at a resolution of 120,000, normalized automatic gain control (AGC) of 300% and RF lens of 40%. Two seconds of maximum cycling time was used to control the number of precursors for tandem-MS/MS (MS2) analysis. In the MS experiments, charge states included 2−6 charges. Dynamic exclusion was used to exclude the previously selected precursors for 35 s. MS2 scans were performed at a resolution of 15,000 (at m/z 200), with

the AGC target value set to "auto". The isolation window was 1.4 m/z and HCD fragmentation was induced with 30 normalized collision energy (NCE). Isotopes were not used for MS2 analysis.

The raw data resulting from the MS runs were searched against the Arabidopsis thaliana UniProt FASTA data using FragPipe (version 18); label-free quantification was performed using LFQ-MBR workflow[98]. Proteins from contaminants and decoys were removed from the results. Furthermore, only proteins that were quantified in at least one replicate in each group were kept for further analysis. The peptide enrichments were analyzed using Perseus[99] and presented as a volcano plot. The log2 fold change of peptide intensity is depicted on the x-axis of the volcano plots, and the -log10 *P*-value of the difference between the two data sets is shown on the y-axis. These differences were calculated using the Student's *t*-test, moderated by Benjamini−Hochberg's method. FDR was set to 0.05 or 0.1 and s0 was set to 0.1–2.

## 60S Ribosome extraction
Polysome extraction was performed based on Hsu et al.[100] but without detergent in the extraction buffer to obtain the cytosolic polysome fractionation. Briefly, about 50 μL of fresh callus tissue from the control and inducible CRISPR/Cas9 mutant lines six days after Cas9 induction were extracted in 500 μL of polysome extraction buffer (100 mM Tris·HCl (pH 8), 40 mM KCl, 20 mM MgCl2, 1 mM DTT, 100 μg/mL cycloheximide, and 10 units/mL DNase I). After keeping on ice for 10 minutes, the debris was removed from the lysate by centrifuging at 16,000 × g for 15 minutes at 4 °C in a JA-25.50 rotor and Avanti J-20 XP centrifuge (Beckman Coulter). About 450 μL of the clear supernatant was then gently loaded on a 15% to 60% sucrose gradient, properly balanced, and centrifuged at 50,000 RPM in an SW55.1 rotor and L8-M ultracentrifuge (Beckman Coulter). The gradient samples were then fractionated using an ISCO absorbance detector (model # UA-5, ISCO, Lincoln, NE). Fifteen ribosome-containing fractions of ~300 μL were obtained after fractionation. RNA was extracted from individual fractions using Trizol reagent. The fractions containing the 60S ribosome subunit were identified by loading the RNA on a 15% TBE-Urea polyacrylamide gel, running at 200 V for 80 min, followed by imaging with a blue light transilluminator and visualize the fraction(s) containing mainly 25S rRNA.

## FIB-SEM
Pieces of Arabidopsis root tips were high pressure frozen with an HPM100 (Leica microsystems, Wetzlar, Germany), 200 μm side of golden carriers and 0.15 M sucrose (made with tap water) as a cryo protectant. Samples were further processed by freeze substitution where the carriers were loaded to a frozen cocktail of 1% OsO4, 0.5% Uranyl acetate and 5% boiled deionized water in acetone. The AFS2 (Leica microsystems, Wetzlar, Germany) was set to a linear warm up of 5 °C/h, starting at −90 °C, with a final 1 h incubation at 20 °C. The samples were washed in acetone, infiltrated with Durcupan resin (Merck Life Science, Solna, Sweden) and polymerized at 65 °C.

For FIB-SEM a small cube of the samples was mounted on a 5 mm aluminum stub with super and silver glue and sectioned to reach a flat surface and the center of the root. The sample was further sputter-coated with a 5 nm layer of Pt to enhance conductivity. A 700 nm-1 μm Platinum layer was added to the area of interest and a trench was milled around it to expose the imaging surface. All volumes were acquired using a Scios DualBeam (FEI, Eindhoven, The Netherlands) and the Auto slice and view 4 (Thermo Fisher Scientific, v.4.0) with the electron beam operating at 2 kV and 0.2 nA and the T1 backscattered electron detector.

The volume was further registered and processed by ImageJ/Fiji software using the plugins Linear alignment by SIFT and Multistackreg. After registration the volumes were converted to mrc-format and the header was modified to add the pixel size. All modeling was done using the Imod software package v.4.9.13(10.1006/jsbi.1996.0013). Sequential SEM images were obtained at a 30 nm interval. And the contour of each mitochondrion on each slice was labeled and a serial of contours of the same mitochondrion were reconstructed into a 3D shape. Different mitochondria were classified if they are not fused, and the mitochondrial matrix is not connected. And different mitochondria were marked with different colors.

## Statistics and Reproducibility
No data was excluded from the analyses. All statistical analyses were conducted using Excel. The specific parameters are provided in connection to each dataset. Representative images for microscopy and immunoblotting are shown from at least 3 biological independent replicates.

## Reporting summary
Further information on research design is available in the Nature Portfolio Reporting Summary linked to this article.

## Data availability
All data of this study are available in the main text, the supplementary data and the source data are provided with this paper. The original proteomics data has been submitted to ProteomeXchange via the PRIDE database with the identifier PXD057599. The proteomics data generated in this study have been deposited in the ProteomeXchange via the PRIDE database under accession code PXD057599.

The genes and mutants used in this study are: *OPNR* (AT5G43822); *opnr-1* (SALK_148287) *OAP1* (AT5G07950); *oap1* (SALK_042474); *OAP2* (AT3G49645); *CDC48D* (AT2G03670); *CIP111* (AT3G56690); *RPL24C* (AT2G44860); *NOG1-1* (AT1G50920) and *BUD20* (AT2G36930). Source data are provided with this paper.

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

## Acknowledgements

We would like to thank Dr. Sara Henriksson from Umeå Centre for Electron Microscopy (UCEM) at the Chemical Biological Centre (KBC), Umeå University, National Microscopy Infrastructure for her help with FIB-SEM experiments (VR-RFI 2019-00217). We thank the Umeå University Protein Expertise Platform (PEP) for help with the recombinant protein expression. We thank Dr. Daria Chrobok at DC SciArt (www.dariasciart.com/) for the design of the schema in Fig. 6. This work was supported by grants from the Swedish Research Council (VR grant 2019-03717, T.N.), Bio4Energy (Swedish Programme for Renewable Energy), and the Knut and Alice Wallenberg Foundation (KAW 2016.0352 and KAW 2020.0240).

## Author contributions

W.W. conducted the experiments, A. Mahboubi and J.H. conducted the 60S ribosome extraction experiments and data analysis, S.Z. and A. Mateus conducted the mass spectrometry analysis, W.W. and T.N. designed the experiments, analyzed the data and wrote the paper with contributions from all authors.

## Funding

## Competing interests

The authors declare no competing interests.
