## [Transparent Peer Review file · Nature Communications]

Ribosome biogenesis in plants requires the nuclear envelope and mitochondria localized OPENER complex

Corresponding Author: Professor Totte Niittylä

Version 0:

Reviewer comments:

Reviewer #1

(Remarks to the Author)

This study analyzed the OPA1 and OAP2 identified for interacting with OPNR. Loss of function of either gene impairs male and female gametophyte development. To gain information on these two proteins, the authors conducted Co-IP analysis and identified AAA+ ATPases CDC48D and CIP111. Mutation of these two genes also causes male and female gametophyte development defects. All these genes are highly expressed in actively dividing cells. Interestingly, OAP1, CDC48D, and CIP111 co-localize with OPNR on the nuclear envelope and the periphery of mitochondria. Using co-IP, TurboID labeling, GFP-Trap Co-IP, AlphaFold, and DeepMSA2-based protein folding (DMFold), the authors established that OPNR, OAP1, OAP2, CDC48D, and CIP111 form an AAA+ ATPase complex, where OPNR, OAP1, and OAP2 form a trimer with 1:1:1 ratio. Furthermore, based on studies in yeast and humans, the authors hypothesized that plant ribosome biogenesis factor RPL24C is the substrate of the OPNR complex, where the complex functions to extract RPL24C from the pre-60S ribosome exported from the nucleus. Interaction of the complex with RPL24C was indicated by protein interaction assays and the AlphaFold algorithm. The authors also detected enrichment of RPL24C in the 60S ribosome when either OAP1 or CDC48D function is compromised. The authors conclude that OPNR, OAP1, OAP2, CDC48D, and CIP111 form a complex at the nuclear membrane. The complex functions to extract RPL24C from the pre-60S ribosome exiting from the nucleus. The complex may also connect to mitochondria, presumably facilitating protein trafficking to mitochondria. The experiments are elegantly designed, overcoming several challenging obstacles. Results are of high standards. The findings reveal the function of these proteins and provide insights into a conserved mechanism of 60S ribosome maturation. The manuscript was well written. I only have a few suggestions for the authors to consider.

1. Only one allele for each gene? A complemented line can be included in the analyses.
2. The mScarlet and YFP signals demonstrated overlap in the cytosol (Figure 3D), but not uniform. Any explanation for that?
3. Figure S2: A-D; More pollens should be included in the staining and germination images. F: pollen grains missed p. Also add the germination data for the cip111 mutant pollens.
4. Figure 3: OPNR-YFP patterns in C and D are different. Explanation should be included. Also, for some reason, the middle image in E is black. G-I: Different mitochondria were marked with different colors. It helps readers understand what the classification is based on.
5. Figure S3: Information of mit-GFP, NUP64-CFP, SUN2-CFP, PHB3-mCherry, etc, should be provided in the figure legend.
6. Figure 5: Only the trimer can pull down RPL24C? If possible, show individual protein pull-down results. In addition, other approaches should be used to verify the interaction. The middle images in D and F are black.
7. Line 186: the subtitle goes, "OAP1, OAP2, CDC48D and CIP111 are predominantly expressed in meristematic tissues". However, no data for OAP2 were provided.
8. Figure S6A. If a chromatograph showing the proteins before His-tag removal is provided, it strengthens the conclusion.

Reviewer #2

(Remarks to the Author)

The manuscript by Wang et al., presents a follow-up story on the role of the Arabidopsis gene OPENER (OPNR) in ribosome biogenesis. The authors identified the OPNR complex, which comprises OPNR, OAP1, OAP2, CDC48D, and CIP111,

through IP-MS. By using genetic and cytological approaches, they demonstrated the critical role of the OPNR complex in both gametophyte and vegetative development, along with notable findings regarding their subcellular localization. The authors find that the defects of OPNR complex leads to the cytosolic retention of NOG1, BUD20, and RPL24C, indicating defective ribosome biogenesis, they proved that the OPNR complex is essential for the extraction of RPL24C from pre-60S ribosomes.

The authors discovered some interesting findings and provide solid experiments to prove their results. However, some aspects still need to be explored, I suggest the following changes and additions to improve this manuscript:

Major points:

- 1) In Figure 4 and Figure 5, the biochemical characterizations of OPNR complex are mainly established from yeast and human studies based on sequence or structural similarities, then the author revealed a similar mechanism with humans, it seems that the novelty of its underlying mechanism is restricted. This reviewer suggests the author to clarify their novel contributions within this context or gain deeper into the mechanism.
- 2) For Figure 5K, the polysome profile pattern is similar for polysome fractions in these opnr complex mutants (all of them are decreased), but they exhibit differently for 40S and 60S fractions. As the author demonstrate the OPNR complex functions together in 60S ribosome assembly, why do they exhibit different patterns on 40S and 60S fractions?

Minor points:

- 1) Line 186, the author demonstrates "OAP1, OAP2, CDC48D and CIP111 are predominantly expressed in meristematic tissues", but OAP2 results was not found in this paragraph.
- 2) In the paragraph starting at line 225, the author notes that CDC48D expression affect the nuclear envelope distribution of OPNR, OAP and CIP111. So how this phenomenon is related to the ribosome biogenesis function?
- 3) Some typo for OAP1 and OAP2, for example, the legend title for Figure S3 is "OAP2, CDC48D and CIP111 subcellular localization", but OAP2 was not found in these figures. In line 254, CDC48D is mentioned to affect the distribution of OAP2, but the above experiment demonstrates OAP1 (Line 248).
- 4) Line 321, the authors indicate that interactions between CDC48D and CIP111 were predicted using DMFold structures. However, there is a lack of experimental validation to confirm these direct interactions.
- 5) Line 414, the author proves that OPNR complex is essential for release of NOG1-1 and BUD20. But the subcellular distribution changes are very different for NOG1-1 and BUD20 in Figure 5F and 5H, for example, in icr-opnr mutants, YFP-NOG1-1 is still dominantly localized in nucleus, but BUD20-YFP is predominantly localized in cytoplasm. Could the author comment on these differences?
- 6) In this paper, the author investigated many different characters for OPNR, OAP1, OAP2, CDC48D, and CIP111 proteins. The author selects different combinations to perform the experiments. For example, in Figure 5F/G the author uses icr-opnr mutants; but in Figure 5I/J the author uses icr-oap1 and icr-cdc48D; then in Figure 5K it appears icr-oap2 and lacks icr-opnr mutants. Since OPNR is the central point of the paper, so how about the polysome profile pattern of icr-opnr mutants. I suggest it would be better to keep consistent for the same conclusions.

Reviewer #3

(Remarks to the Author)

The article by Wang et al. provides a thorough investigation into the role of two OPENER-ASSOCIATED proteins (OAPs), an area that has not yet been fully characterized. Through a combination of microscopy, proteomics, genetics, and CRISPR-based techniques, the authors create mutant plants and methodically explore the functions of these proteins. The study builds upon the authors' previous work, presenting a novel function of the complex they initially discovered, which has not been extensively studied until now.

A key strength of the study is the robust proteomics analysis, including in vivo immunoprecipitation (IP) and AlphaFold models, which provide strong evidence of interactions between OPNR, OAP proteins, and CIP111. These findings are well-supported by complementary microscopy data, adding depth to the overall analysis. Proving protein complexes is a particularly demanding task, and this work is commendable for its careful and thorough approach. Overall, the study by Wang et al. is well-executed. However, the microscopy data require further refinement, particularly through colocalization analysis and the use of advanced assays. Additionally, including a summary cartoon or graphical abstract would significantly enhance the readability and accessibility of the study.

Specific Points:

While the study provides robust evidence for OPNR complex formation, additional work may be required to fully address other dimensions implied by the title. The reader may feel that the majority of the results relate to the composition and localization of the complex itself, rather than its role in ribosome metabolism. Expanding the scope to explore these aspects in greater detail could strengthen the alignment between the title and the presented findings, particularly because several conclusions are based on functional homology.

- A) Given the extensive proteomics work undertaken, it would be useful to include notes on the ribosomal proteins identified in the study. For instance, how do the authors explain the absence of RPL24 from the shortlist of candidate proteins?
- B) Since the hypothesis suggests shuffling of ribosomal proteins across different cellular compartments, the results currently appear somewhat static. For example:

The authors might consider incorporating FRAP (fluorescence recovery after photobleaching) experiments to examine the exchange rate of ribosome assembly factors and complex members between the cytoplasm and nucleus.

Additionally, it would be valuable to test whether ribosome assembly factors can move to the cytoplasm in the mutant background.

- C) Is the reduction in rRNA transcript levels expected? Or are the observed defects solely linked to ribosome dysfunction?
- D) What is the connection between ribosome defects and the observed growth defects? Does this affect the speed of protein synthesis, protein folding, or some other aspect of ribosome function? Could certain specific proteins be disproportionately affected? These points should be addressed.
- E) Mitochondrial Localization and Function

Regarding the mitochondrial localization of this complex, is it expected to play a role in mitoribosome biogenesis?

Additional Comments:

Line 85: The claim about the role of OPNR in ribosome biogenesis lacks a citation. Adding an appropriate reference here is essential to support the statement.

Panel 3 and Supplementary Figure S3: The microscopy analysis in these figures requires more rigorous colocalization analysis, particularly regarding the mitochondrial patterns ("spots") and other relevant panels.

Panel 3G–I: The description of these panels is unclear. It is not evident what specific data is being evaluated, and there is no reference to raw FIB-SEM data in the main text. It seems like this data is from a previous paper. Including the reference and providing additional details will improve transparency and reproducibility. Also, what is the relevance of increased mitochondrial contacts?

Lines 238–239: A more robust approach, such as Förster Resonance Energy Transfer (FRET)-FLIM, could be implemented to quantitatively evaluate the distance between the proteins and confirm their interaction potential.

Lines 253–255: The explanation provided in these lines is not further discussed. The authors should clarify how the presented results link to the interaction potential of the proteins. A more detailed discussion would bridge the gap between the experimental findings and their interpretation.

Lines 476–481: This section is interesting but may not be directly relevant to the main focus of the paper.

Lines 490–494: Please clarify this section. While ribosome biogenesis is important overall, if there are specific functions linked to gametophyte development, they should be mentioned explicitly.

Line 566: Please verify the figure numbers for accuracy.

Line 589: Please confirm whether "data not shown" is allowed under the journal's policy.

Figure S2F: Please check the spelling on the Y-axis.

Supplemental Data: The supplemental data and raw data are not properly referenced. For example, the meaning of "Dataset S1" and "S1C" is unclear. I only found one Excel file, and the individual sheets were not labeled accordingly.

Version 1:

Reviewer comments:

Reviewer #1

(Remarks to the Author)

The revised manuscript has addressed all my concerns. Hence, I recommend acceptance for publication in Nature Communications.

Reviewer #2

(Remarks to the Author)

In this new version, authors have addressed all my questions and concerns. I have no additional concerns.

Reviewer #3

(Remarks to the Author)

Wang and colleagues have submitted a revised manuscript that thoroughly addresses most of the reviewers' comments. However, a few points still require attention:

Following on the previous comment (Rev#3 A) and given answer, it would be very useful to examine the interaction potential of RPL24 and OPENER/OAP using BiFC (since authors have that in hands), particularly given that BiFC is a robust system capable that may be reveal those transient interactions.

I still think the representative FIB-SEM image, or link to the raw data should be provided.

The nomenclature should be consistent throughout the manuscript (Arabidopsis vs yeast). For example, gene/protein names such as Cell Division Cycle 48 D (CDC48D), Calmodulin-interacting protein 111 (CIP111), and Ribosomal-like protein 24 (RPL24) I guess it should follow the same formatting.

In Supplementary Figure S2F,G, the "%" symbol is unnecessary next to each number, as the figure title already states that values are expressed in percentages.

In my opinion, the responses to the first and sixth comments from Reviewer 2 should be briefly summarized and incorporated into the main text of the manuscript, rather than remaining solely in the rebuttal.

Please review and correct the citation formatting and content throughout the manuscript—for instance, citation (46) and others may require revision.

Dear Editors,

We thank the editors and reviewers for the recognition of the importance of the work, and for the constructive comments that allowed us to improve the manuscript. Please find below our responses to the reviewer's comments.

REVIEWER COMMENTS

Reviewer #1 (Remarks to the Author):

This study analyzed the OPA1 and OAP2 identified for interacting with OPNR. Loss of function of either gene impairs male and female gametophyte development. To gain information on these two proteins, the authors conducted Co-IP analysis and identified AAA+ ATPases CDC48D and CIP111. Mutation of these two genes also causes male and female gametophyte development defects. All these genes are highly expressed in actively dividing cells. Interestingly, OAP1, CDC48D, and CIP111 co-localize with OPNR on the nuclear envelope and the periphery of mitochondria. Using co-IP, TurboID labeling, GFP-Trap Co-IP, AlphaFold, and DeepMSA2-based protein folding (DMFold), the authors established that OPNR, OAP1, OAP2, CDC48D, and CIP111 form an AAA+ ATPase complex, where OPNR, OAP1, and OAP2 form a trimer with 1:1:1 ratio. Furthermore, based on studies in yeast and humans, the authors hypothesized that plant ribosome biogenesis factor RPL24C is the substrate of the OPNR complex, where the complex functions to extract RPL24C from the pre-60S ribosome exported from the nucleus. Interaction of the complex with RPL24C was indicated by protein interaction assays and the AlphaFold algorithm. The authors also detected enrichment of RPL24C in the 60S ribosome when either OAP1 or CDC48D function is compromised. The authors conclude that OPNR, OAP1, OAP2, CDC48D, and CIP111 form a complex at the nuclear membrane. The complex functions to extract RPL24C from the pre-60S ribosome exiting from the nucleus. The complex may also connect to mitochondria, presumably facilitating protein trafficking to mitochondria. The experiments are elegantly designed, overcoming several challenging obstacles. Results are of high standards. The findings reveal the function of these proteins and provide insights into a conserved mechanism of 60S ribosome maturation. The manuscript was well written. I only have a few suggestions for the authors to consider.

1. Only one allele for each gene? A complemented line can be included in the analyses.

RE: All the mutant lines were complemented to confirm the causal relationship with the phenotype. The complementation results are shown in Fig. 1A. *oap1*; *OAP1-mNG* is *oap1* mutant complemented with *pOAP1:OAP1-mNeonGreen*. *oap2*; *pOAP2:OAP2* is *oap2* mutant complemented with *pOAP2:OAP2*. *cip111*; *mScar-CIP111m* is *cip111* mutant complemented with *pCIP111:mScarlet-CIP111m*. *cdc48D*; *mScar-CDC48D* is *cdc48D* mutant complemented with *pCDC48D:mScarlet-CDC48D*. To clarify these results a description of each complementation line was added to the figure legend. The

complementation experiments are also described in the main text on lines 114-117, 150-153, and 164-168.

2. The mScarlet and YFP signals demonstrated overlap in the cytosol (Figure 3D), but not uniform. Any explanation for that?

RE: mScarlet-CDC48D was shown to partially colocalized with the mitochondria marker mit-GFP (Supplementary Fig. 3C), and in our previous publication¹ OPNR-YFP was also shown to localize to mitochondria. The OPNR-YFP signals overlap with mScarlet-CDC48D shown in Fig. 3D is not uniform possibly because not all the OPNR proteins are interacting with mScarlet-CDC48D at the same time. We added a discussion part on lines 676-687.

3. Figure S2: A-D; More pollens should be included in the staining and germination images. F: pollen grains missed p. Also add the germination data for the *cip111* mutant pollens.

RE: Supplementary Fig. 2A-E were changed to images showing more pollen grains. A histogram was also added as Supplementary Fig. 2F to show the ratio of normal and abnormal pollen grains. The text on lines 178-183 was revised accordingly. The number of pollen grains was also listed in the figure legends of Supplementary Fig. 2F, G. The spelling of pollen was corrected in Supplementary Fig. 2G.

The reason that the germination data for the *cip111* mutant pollen was not included is that the viability staining results indicated that the mutant pollen is dead. The expectation is that ~25% of the pollen grains from *cip111/+; mScarlet-CIP111m/-* carry the *cip111* mutation but not the *mScarlet-CIP111m* transgene. Thus, the observation that 23.1% of the pollen grains stained green (Supplementary Fig. 2C, F) indicated that the *cip111* mutant pollen grains were not viable. Therefore, the *in vitro* pollen germination assay was not performed for *cip111/+; mScarlet-CIP111m/-*.

4. Figure 3: OPNR-YFP patterns in C and D are different. Explanation should be included. Also, for some reason, the middle image in E is black. G-I: Different mitochondria were marked with different colors. It helps readers understand what the classification is based on.

RE: The different patterns of OPNR-YFP in Fig.3C, D are explained on lines 241-256. OPNR, OAP1 and CIP111 show similar localization in the cell (Fig. 3). We observed that the expression of N-terminal fusions of mScarlet- or YFP-CDC48D affect the distribution of CDC48D itself and the other components of the OPENER complex.

pRPS5A:XVE is a β -estradiol inducible promoter. The middle image in Fig. 3E, F are YFP channel images of root tip cells from seedlings harboring *pRPS5A:XVE:YFP-CDC48D* before inducing (Fig. 3E) and after inducing (Fig. 3F). The settings of the YFP

channel in the middle image of Fig. 3E, F are the same. Before inducing, the YFP channel gave no signal from YFP-CDC48D (Fig. 3E), while after inducing there was a signal of YFP-CDC48D (Fig. 3F). To clarify this aspect a description was added in the legends of Fig. 3E. and supplementary Fig. 3G.

Fig. 3G-I show separate mitochondria with different colors for clarity. Mitochondria were classified separate if they were not fused, and the mitochondrial matrix was not connected. Details of the classification are now included in the methods under FIB-SEM, and also the legend of Fig. 3G-I was modified accordingly.

5. Figure S3: Information of mit-GFP, NUP64-CFP, SUN2-CFP, PHB3-mCherry, etc, should be provided in the figure legend.

RE: The information of mt-CFP, mit-GFP, NUP64-CFP, SUN2-CFP and PHB3-mCherry was added in the legend of Supplementary Fig. 3, and is also found on the lines 233-238 and 261-263 of the main text.

6. Figure 5: Only the trimer can pull down RPL24C? If possible, show individual protein pull-down results. In addition, other approaches should be used to verify the interaction. The middle images in D and F are black.

RE: The AlphaFold modeling indicated that the C-terminus of RPL24C is placed into the pocket formed by the OPNR-OAP1-OAP2 trimer. Despite multiple expression attempts the individual proteins of OPNR, OAP1 and OAP2 exhibited limited solubility when expressed alone in *E. coli* cells. In contrast the OPNR-OAP1-OAP2 trimer was soluble and could be purified from the supernatant after cell lysis and centrifugation steps. To achieve this, it was necessary to clone OPNR, OAP1 and OAP2 into the same vector to co-express them in the same cell. This observation indicated that individual proteins of OPNR, OAP1 and OAP2 were not folding properly in the *E. coli* cells. Thus, individual protein pull-down assays could not be performed between OPNR, OAP1, OAP2 and RPL24C.

The requirement of all four proteins (OPNR, OAP1, OAP2 and RPL24C) and maybe other conditions make the interaction verification using other conventional approaches like yeast 2-hybrid unsuitable or at least challenging to use. More advanced fluorescence based *in vivo* methods like FLIM-FRET could be an option. However, as also discussed below in response to reviewer 3, these experiments would require new transgenic lines and take significant time to prepare. Furthermore, it is not certain that the fluorophores fused to OPNR complex and RPL24C would have the appropriate relative orientation and distance for a FRET assay to work.

The interaction between the OPNR complex and RPL24C in pre-60S ribosome should be transient and likely tightly regulated *in vivo*, which may explain why we did not catch RPL24C in our Co-IP-MS experiments with the OPNR complex. However, we would

also like to emphasize that the RPL24C was identified in the TurboID proximity labeling of CDC48D neighboring proteins as shown in Supplementary Fig. 4B. TurboID proximity labeling can catch transient interactions *in vivo* which can explain RPL24C identification in this experiment in line with the proposed mode of action of OPNR complex.

The middle images of lower panels of Fig 5D, F and H did not show mScarlet signal. The black panels illustrate successful inducible expression of CRISPR/Cas9 and targeting of the OPNR-mScarlet transgene. Before inducing, the middle images of the upper panels show OPNR-mScarlet signals. While after CRISPR/Cas9 induction, the OPNR-mScarlet signal is reduced as shown by the almost black images obtained with the same setting of the mScarlet channel as the upper panels. To clarify this aspect a description was added in the figure legend of Fig. 5D.

7. Line 186: the subtitle goes, "OAP1, OAP2, CDC48D and CIP111 are predominantly expressed in meristematic tissues". However, no data for OAP2 were provided.

RE: Thank you for pointing out this mistake. The *pOAP2:OAP2-YFP* and *pOAP2:YFP-OAP2* cannot complement the *oap2* mutant phenotype and did not show any YFP signal. Thus, the OAP2 was removed from the subtitle on **line187**.

8. Figure S6A. If a chromatograph showing the proteins before His-tag removal is provided, it strengthens the conclusion.

RE: We run the proteins before and after His-tag removal again and the chromatographs are now included in Supplementary Fig. 6A. Also please note that the His-OAP1-OAP2-OPNR trimer proteins before His-tag removal were used for the pull-down experiments with GST-RPL24C in Fig. 5C. The western-blot with anti-His-tag antibody showed the correct size of His-OAP1.

Reviewer #2 (Remarks to the Author):

The manuscript by Wang et al., presents a follow-up story on the role of the Arabidopsis gene OPENER (OPNR) in ribosome biogenesis. The authors identified the OPNR complex, which comprises OPNR, OAP1, OAP2, CDC48D, and CIP111, through IP-MS. By using genetic and cytological approaches, they demonstrated the critical role of the OPNR complex in both gametophyte and vegetative development, along with notable findings regarding their subcellular localization. The authors find that the defects of OPNR complex leads to the cytosolic retention of NOG1, BUD20, and RPL24C, indicating defective ribosome biogenesis, they proved that the OPNR complex is essential for the extraction of RPL24C from pre-60S ribosomes.

The authors discovered some interesting findings and provide solid experiments to prove their results. However, some aspects still need to be explored, I suggest the

following changes and additions to improve this manuscript:

Major points:

1) In Figure 4 and Figure 5, the biochemical characterizations of OPNR complex are mainly established from yeast and human studies based on sequence or structural similarities, then the author revealed a similar mechanism with humans, it seems that the novelty of its underlying mechanism is restricted. This reviewer suggests the author to clarify their novel contributions within this context or gain deeper into the mechanism.

RE: In comparison to yeast our results reveal increased pre-ribosome processing complexity in plants, and although the OPNR complex has similarities to its human counterpart, its subcellular localization on both nuclear envelope and mitochondria is a novel finding. Especially the mitochondria localization of the OPNR complex and the defective mitochondria morphology in the mutants of the OPNR complex point to previously unrecognized subcellular complexity of ribosome biogenesis in plants and mitochondria association as a feature to ensure sufficient translational capacity. Our work emphasizes the importance of studying ribosome biogenesis in different organisms. The overall process appears conserved in eukaryotes, but our results reveal differences that warrant further investigation. In a wider context, ribosomes are important determinants of crop yield and performance, and our work provides a new opening and tools to study this important topic.

2) For Figure 5K, the polysome profile pattern is similar for polysome fractions in these *opnr* complex mutants (all of them are decreased), but they exhibit differently for 40S and 60S fractions. As the author demonstrate the OPNR complex functions together in 60S ribosome assembly, why do they exhibit different patterns on 40S and 60S fractions?

RE: Although the OPNR complex functions together, the phenotypes are not identical between the mutants of different complex components. The phenotypes of *opnr*, *oap1* and *oap2* are similar to each other as shown in Fig. 1A, C: Ovule and seed development was impaired in *opnr*, *oap1* and *oap2* mutants at a similar stage. The mutants of CDC48D and CIP111 showed more severe phenotype: both ovule and pollen development are affected. The *cip111* shows an even stronger phenotype than *cdc48D*; pollen and ovule development in *cip111* was arrested at earlier stages than *cdc48D* as shown in Fig. 1 and Supplementary Fig. 2. Thus, CDC48D and CIP111 may have additional cofactors and/or substrates, this possibility is discussed on lines 470-490.

In addition, the samples analyzed in Figure 5K were extracted from inducible CRISPR/Cas9 mutants. The efficiency of inducible CRISPR/Cas9 knockout of different components of the OPNR complex is different as shown in Fig. 2C. Taken together, the differences in mutant phenotypes and the CRISPR/cas9 efficiency may both contribute to differences in 40S and 60S fractions. Furthermore, it seems possible that

a primary effect on 60S assembly could also lead to changes in the 40S fraction. We added this part to the discussion on **lines 688-699**.

Minor points:

1) Line 186, the author demonstrates “OAP1, OAP2, CDC48D and CIP111 are predominantly expressed in meristematic tissues”, but OAP2 results was not found in this paragraph.

RE: Thank you for pointing this out. *pOAP2:OAP2-YFP* and *pOAP2:YFP-OAP2* cannot complement the *oap2* mutant phenotype and did not show YFP signals. OAP2 was removed from the subtitle.

2) In the paragraph starting at line 225, the author notes that CDC48D expression affect the nuclear envelope distribution of OPNR, OAP and CIP111. So how this phenomenon is related to the ribosome biogenesis function?

RE: On **lines 241-256** we describe how OPNR, OAP1 and CIP111 all show similar localization on nuclear envelope and mitochondria in the presence of the native CDC48D proteins. Since OPNR, OAP1, OAP2, CDC48D and CIP111 form a complex it follows that the native CDC48D should also localize to nuclear envelope and mitochondria. The N-terminal YFP or mScarlet fusion of CDC48D affected the distribution of OPNR, OAP1 and CIP111. The nuclear envelope localization was not clear, while the mitochondria localization is largely unaffected. Thus, it seems that the N-terminal fusion of YFP or mScarlet altered the distribution of CDC48D and associated proteins for some reason. Interestingly we observed that the mScarlet-CDC48D fusion protein could complement the ovule arrest phenotype of *cdc48D* as shown in Fig 1A, which indicates that maybe the mitochondria associated 60S ribosome assembly with the mitochondria localized OPNR complex is sufficient during embryo and early seedling development. We added this part to the discussion on **lines 676-687**.

3) Some typo for OAP1 and OAP2, for example, the legend title for Figure S3 is “OAP2, CDC48D and CIP111 subcellular localization”, but OAP2 was not found in these figures. On line 254, CDC48D is mentioned to affect the distribution of OAP2, but the above experiment demonstrates OAP1 (Line 248).

RE: Thank you for pointing this out. The legend title for Figure S3 was changed to “OAP1, CDC48D and CIP111 subcellular localization”. OAP2 was changed to OAP1 on **lines 254-255**.

4) Line 321, the authors indicate that interactions between CDC48D and CIP111 were predicted using DMFold structures. However, there is a lack of experimental validation to confirm these direct interactions.

RE: The co-IP and TurboID proximity labeling experiments (Supplementary Fig. 4 and 5) show that the CDC48D and CIP111 are part of the same complex. Both proteins are AAA-ATPases which have been shown to form heteromers ², and this together with the modelling supports direct interaction. Besides, the localization of YFP-CIP111 on nuclear envelope was affected by the co-expression of mScarlet-CDC48D (Supplementary Fig. 3I) and N-terminal fusion of mScarlet may affect the localization of CDC48D as discussed on lines 676-687. This observation also suggested that CDC48D interacts with CIP111 *in vivo*.

In addition, we carried out a low background bimolecular fluorescence complementation (BiFC) assay ³ which provided additional experimental validation of the CDC48D and CIP111 interaction *in vivo*. These results including the control data are now included on lines 341-346 and in Supplementary Fig. 6J.

5) Line 414, the author proves that OPNR complex is essential for release of NOG1-1 and BUD20. But the subcellular distribution changes are very different for NOG1-1 and BUD20 in Figure 5F and 5H, for example, in *icr-opnr* mutants, YFP-NOG1-1 is still dominantly localized in nucleus, but BUD20-YFP is predominantly localized in cytoplasm. Could the author comment on these differences?

RE: It is clear that NOG1-1 and BUD20 show different subcellular distribution after mutation of the OPNR complex components (Fig. 5F, H and Supplementary Fig. 7F, G). In yeast study, although Nog1 and Bud20 are cofactors for 60S ribosome assembly, Nog1 joins the pre-60S ribosome early in the nucleolus, while Bud20 joins pre-60S later in the nucleoplasm ⁴. The knockout of Nog1 in yeast is lethal ⁵, while Bud20 knockout only shows a slow-growth phenotype ⁶. Extrapolating from the yeast observations it is possible that NOG1-1 and BUD20 in Arabidopsis also play different roles in pre-60S ribosome assembly and behave differently in response to knockout of the OPNR complex components. Furthermore, it is also possible or even likely that NOG1-1 and BUD20 have different synthesis and turnover rates and show different localization in the *icr-opnr* and *icr-cdc48D* mutants. We added these possibilities to the discussion on lines 602-612.

6) In this paper, the author investigated many different characters for OPNR, OAP1, OAP2, CDC48D, and CIP111 proteins. The author selects different combinations to perform the experiments. For example, in Figure 5F/G the author uses *icr-opnr* mutants; but in Figure 5I/J the author uses *icr-oap1* and *icr-cdc48D*; then in Figure 5K it appears *icr-oap2* and lacks *icr-opnr* mutants. Since OPNR is the central point of the paper, so how about the polysome profile pattern of *icr-opnr* mutants. I suggest it would be better to keep consistent for the same conclusions.

RE: Since OPNR, OAP1 and OAP2 function as a heterotrimer and the corresponding mutants show almost identical phenotypes in ovule and seed development, the *icr-oap2* based conclusions are likely to be representative of the heterotrimer defects.

Reviewer #3 (Remarks to the Author):

The article by Wang et al. provides a thorough investigation into the role of two OPNER-ASSOCIATED proteins (OAPs), an area that has not yet been fully characterized. Through a combination of microscopy, proteomics, genetics, and CRISPR-based techniques, the authors create mutant plants and methodically explore the functions of these proteins. The study builds upon the authors' previous work, presenting a novel function of the complex they initially discovered, which has not been extensively studied until now.

A key strength of the study is the robust proteomics analysis, including in vivo immunoprecipitation (IP) and AlphaFold models, which provide strong evidence of interactions between OPNR, OAP proteins, and CIP111. These findings are well-supported by complementary microscopy data, adding depth to the overall analysis. Proving protein complexes is a particularly demanding task, and this work is commendable for its careful and thorough approach. Overall, the study by Wang et al. is well-executed. However, the microscopy data require further refinement, particularly through colocalization analysis and the use of advanced assays. Additionally, including a summary cartoon or graphical abstract would significantly enhance the readability and accessibility of the study.

RE: We agree and have included a graphical abstract in Fig. 6 to enhance the readability.

Specific Points:

While the study provides robust evidence for OPNR complex formation, additional work may be required to fully address other dimensions implied by the title. The reader may feel that the majority of the results relate to the composition and localization of the complex itself, rather than its role in ribosome metabolism. Expanding the scope to explore these aspects in greater detail could strengthen the alignment between the title and the presented findings, particularly because several conclusions are based on functional homology.

A) Given the extensive proteomics work undertaken, it would be useful to include notes on the ribosomal proteins identified in the study. For instance, how do the authors explain the absence of RPL24 from the shortlist of candidate proteins?

RE: The interaction between the OPNR complex and RPL24C can be expected to be transient, and such transient interactions are hard to catch by co-IP/MS. We hypothesize this to be the reason why RPL24C was not in the OPNR complex co-IP/MS identified proteins. This idea is also supported by the identification of RPL24C

(and BUD20) in the TurboID proximity labeling of CDC48D neighboring proteins as shown in Supplementary Fig. 4B.

B) Since the hypothesis suggests shuffling of ribosomal proteins across different cellular compartments, the results currently appear somewhat static. For example: The authors might consider incorporating FRAP (fluorescence recovery after photobleaching) experiments to examine the exchange rate of ribosome assembly factors and complex members between the cytoplasm and nucleus. Additionally, it would be valuable to test whether ribosome assembly factors can move to the cytoplasm in the mutant background.

RE: Homologs of the ribosome assembly factors RPL24C, NOG1 and BUD20 in yeast and humans have been shown to shuttle between the nucleus and cytosol⁵⁻¹². Our results point to similar dynamism in plants: NOG1-1 and BUD20 were localized to the nucleus on the control condition but showed clear cytosolic retention in the mutants of the OPNR complex components (Fig. 5F-H). This indicated that these factors recycled efficiently from cytosol to nucleus in the WT control.

We also added results from new experiments which show that the nuclear export inhibitor leptomycin B causes the retention of RPL24C-YFP, YFP-NOG1-1 and BUD20-YFP in the nucleus (Supplementary Figure 7H to 7J). This observation provides further evidence that these factors are exported from nucleus to cytosol in the wild-type control (lines 425-430). The homologs of the OPNR complexes in yeast and humans were not observed to shuttle between the nucleus and the cytosol⁹⁻¹². Thus, combined with the studies in yeast and humans the results in our study provide evidence of the dynamic shuttle of the ribosome assembly factors between nucleus and cytosol. To clarify this aspect, we added a new part to the discussion on lines 612-621.

C) Is the reduction in rRNA transcript levels expected? Or are the observed defects solely linked to ribosome dysfunction?

RE: Polysome profiling was performed on inducible CRISPR/Cas9 mutant calli and it seems that after CRISPR/Cas9 induction the de novo ribosome biogenesis is impaired which can be expected to lead to reduction of rRNA transcript levels and translation.

D) What is the connection between ribosome defects and the observed growth defects? Does this affect the speed of protein synthesis, protein folding, or some other aspect of ribosome function? Could certain specific proteins be disproportionately affected? These points should be addressed.

RE: It is likely that the growth defects observed in the mutants are caused by general impairment of translation capacity due to defect ribosome biogenesis, which ultimately leads to the observed lethal phenotypes. What is affected is basically the magnitude of translation caused by less available mature and functional ribosomes.

It seems unlikely that there would be any bias in translation of specific proteins, but it is likely that proteins with fast turnover would be disproportionately affected. To investigate the possibility of any translation bias in more detail more experiments like Ribo-seq would be needed. However, we believe such experiments are not within the scope of the current study.

E) Mitochondrial Localization and Function

Regarding the mitochondrial localization of this complex, is it expected to play a role in mitoribosome biogenesis?

RE: Mitoribosome biogenesis has been shown to occur in the matrix of mitochondria¹³. Supplementary Fig. 3J-L suggested that OPNR-YFP, OAP1-YFP and YFP-CDC48D predominantly localize to the periphery of mitochondria. And Supplementary Fig. 4B showed that the mitochondria outer membrane TOM20 proteins were highly ranked in the neighboring proteins of CDC48D together with other components of the OPNR complex, which also indicated that the OPNR complex localizes to the outer surface of mitochondria. Thus, it seems unlikely that the OPNR complex is directly involved in mitoribosome biogenesis. However, mitoribosomal proteins are all encoded by nuclear genes and most of these proteins are suggested to be synthesized on the surface of mitochondria^{13,14}. It is possible that the OPNR complex affects the translation of mitoribosomal proteins on the surface of mitochondria and affects mitoribosome biogenesis indirectly.

Additional Comments:

Line 85: The claim about the role of OPNR in ribosome biogenesis lacks a citation. Adding an appropriate reference here is essential to support the statement.

RE: The sentence begins with “In this study,...” on **line 84** hopefully making it clear that the function of OPNR in ribosome biogenesis is identified in this study. There are no other publications indicating that OPNR is involved in ribosome biogenesis.

Panel 3 and Supplementary Figure S3: The microscopy analysis in these figures requires more rigorous colocalization analysis, particularly regarding the mitochondrial patterns (“spots”) and other relevant panels.

RE: We have now included a quantitative analysis of this data when applicable. The degree of colocalization was determined by the Pearson correlation coefficient (PCC) for data in Fig. 3 and Supplementary Fig. 3. Coloc2 in ImageJ was used.

Panel 3G–I: The description of these panels is unclear. It is not evident what specific data is being evaluated, and there is no reference to raw FIB-SEM data in the main text. It seems like this data is from a previous paper. Including the reference and

providing additional details will improve transparency and reproducibility. Also, what is the relevance of increased mitochondrial contacts?

RE: Fig. 3G-I show FIB-SEM 3D reconstruction of mitochondria. Sequential SEM images were obtained at a 30 nm interval. And the contour of each mitochondrion on each slice was labeled and a serial of contours of the same mitochondrion were reconstructed into a 3D shape. Different mitochondria were classified if they are not fused, and the mitochondrial matrix is not connected. And different mitochondria were marked with different colors. The legend of Fig. 3G-I was modified and more detailed description was added in the methods for FIB-SEM.

Lines 238–239: A more robust approach, such as Förster Resonance Energy Transfer (FRET)-FLIM, could be implemented to quantitatively evaluate the distance between the proteins and confirm their interaction potential.

RE: We agree that validating the interaction by FRET is a good idea, but it is not certain that the fluorophores fused to OAP1, OPNR and CIP111 would be close enough to each other and the FRET would work. We believe that this is something for the follow-up studies and not a critical experiment for the conclusions of the current manuscript.

Lines 253–255: The explanation provided in these lines is not further discussed. The authors should clarify how the presented results link to the interaction potential of the proteins. A more detailed discussion would bridge the gap between the experimental findings and their interpretation.

RE: We have added the following text to the discussion on lines 676-687. “The fluorescent protein fusions of OPNR, OAP1 and CIP111 all show similar localization on the nuclear envelope and mitochondria in the presence of the native CDC48D. It follows that the native CDC48D proteins should also localize to nuclear envelope and mitochondria since OPNR, OAP1, OAP2, CDC48D and CIP111 can form a complex. This conclusion was also supported by the fact that the N-terminal fusion of YFP or mScarlet to CDC48D affected the distribution of OPNR, OAP1 and CIP111 (Fig. 3D-F and Supplementary Fig. 3F-I). In these lines the nuclear envelope localization was not clear anymore, while the mitochondria association was still prominent. Thus, it seems that the N-terminal fusion of YFP or mScarlet altered the distribution of CDC48D and associated proteins for some reason. Interestingly the mScarlet-CDC48D fusion protein could complement the ovule arrest phenotype of *cdc48D* as shown in Fig 1A, which raises the possibility that the mitochondria associated 60S ribosome assembly catalyzed by the OPNR complex can be sufficient. ”

Lines 476–481: This section is interesting but may not be directly relevant to the main focus of the paper.

RE: We believe this section is important as it explains possible reasons for the more

severe phenotype of *cdc48D* and *cip111* than *opnr*, *oap1* and *oap2*. A question also raised by reviewer 2.

Lines 490–494: Please clarify this section. While ribosome biogenesis is important overall, if there are specific functions linked to gametophyte development, they should be mentioned explicitly.

RE: The relationship between gametophyte development, meristem maintenance and ribosome biogenesis were discussed on lines 507-513 as follows “AtNOB1 and AtENP1 are components of the 40S pre-ribosome and function as co-factors for 40S ribosome biogenesis in Arabidopsis. AtNOB1 is an endonuclease cleaving pre-rRNA. Mutants of *AtNOB1* and *AtENP1* show delayed pollen development and reduced transmission of the mutation to the next generation through both female and male gametophytes¹⁵. The AAA-ATPase Midasin 1 (MDN1) is essential for 60S ribosome biogenesis and plays a crucial role in root meristem maintenance. Mutations of MDN1 lead to defects in ribosome assembly and affect root development and meristem maintenance¹⁶. NOTCHLESS (NLE) is involved in 60S ribosome biogenesis and interacts with MDN1. *nle* mutants also show defects in root meristem maintenance and embryo development¹⁷. In this study we also observed pollen defects in *cdc48D* and *cip111* mutants (Supplementary Fig. 2) and the decreased root meristem size in the inducible CRISPR/Cas9 *opnr*, *oap1*, *oap2*, *cdc48D* and *cip111* mutants (Fig. 2B, C). Thus, these phenotypes are consistent with the role of ribosome biogenesis in gametophyte development and meristem maintenance.”

Line 566: Please verify the figure numbers for accuracy.

RE: We have verified and changed the figure numbers on line 594.

Line 589: Please confirm whether "data not shown" is allowed under the journal's policy.

RE: Images related to this data were added in Supplementary Fig. 3M, N.

Figure S2F: Please check the spelling on the Y-axis.

RE: Thank you for pointing this out. We have corrected the spelling of “pollen” in supplementary Fig. 2F.

Supplemental Data: The supplemental data and raw data are not properly referenced. For example, the meaning of “Dataset S1” and “S1C” is unclear. I only found one Excel file, and the individual sheets were not labeled accordingly.

RE: Thank you for pointing this out. We have now labeled the sheets accordingly.

- Mitochondria Localized Protein Required for Cell Cycle Progression in Arabidopsis. *Plant Cell* **31**, 1446–1465 (2019). <https://doi.org/10.1105/tpc.19.00033>
- 2 Krishnamoorthy, V. *et al.* The SPATA5–SPATA5L1 ATPase complex directs replisome proteostasis to ensure genome integrity. *Cell* **187**, 2250–2268 e2231 (2024). <https://doi.org/10.1016/j.cell.2024.03.002>
- 3 Gookin, T. E. & Assmann, S. M. Significant reduction of BiFC non-specific assembly facilitates in planta assessment of heterotrimeric G-protein interactors. *Plant J* **80**, 553–567 (2014). <https://doi.org/10.1111/tpj.12639>
- 4 Altwater, M. *et al.* Targeted proteomics reveals compositional dynamics of 60S pre-ribosomes after nuclear export. *Mol Syst Biol* **8**, 628 (2012). <https://doi.org/10.1038/msb.2012.63>
- 5 Kallstrom, G., Hedges, J. & Johnson, A. The putative GTPases Noglp and Lsglp are required for 60S ribosomal subunit biogenesis and are localized to the nucleus and cytoplasm, respectively. *Mol Cell Biol* **23**, 4344–4355 (2003). <https://doi.org/10.1128/MCB.23.12.4344-4355.2003>
- 6 Bassler, J. *et al.* The Conserved Bud20 Zinc Finger Protein Is a New Component of the Ribosomal 60S Subunit Export Machinery. *Molecular and Cellular Biology* **32**, 4898–4912 (2012). <https://doi.org/10.1128/Mcb.00910-12>
- 7 Klingauf–Nerurkar, P. *et al.* The GTPase Nog1 co-ordinates the assembly, maturation and quality control of distant ribosomal functional centers. *Elife* **9** (2020). <https://doi.org:ARTN> e52474
- 10.7554/eLife.52474
- 8 Durand, S. *et al.* RSL24D1 sustains steady-state ribosome biogenesis and pluripotency translational programs in embryonic stem cells. *Nature Communications* **14** (2023). <https://doi.org:ARTN> 356
- 10.1038/s41467-023-36037-7
- 9 Loibl, M. *et al.* The Drug Diazaborine Blocks Ribosome Biogenesis by Inhibiting the AAA-ATPase Drg1. *Journal of Biological Chemistry* **289**, 3913–3922 (2014). <https://doi.org/10.1074/jbc.M113.536110>
- 10 Kappel, L. *et al.* Rlp24 activates the AAA-ATPase Drg1 to initiate cytoplasmic pre-60S maturation. *J Cell Biol* **199**, 771–782 (2012). <https://doi.org/10.1083/jcb.201205021>
- 11 Pertschy, B. *et al.* Cytoplasmic recycling of 60S preribosomal factors depends on the AAA protein drg1. *Molecular and Cellular Biology* **27**, 6581–6592 (2007). <https://doi.org/10.1128/Mcb.00668-07>
- 12 Ni, C. *et al.* Labeling of heterochronic ribosomes reveals Clorf109 and SPATA5 control a late step in human ribosome assembly. *Cell Rep* **38**, 110597 (2022). <https://doi.org/10.1016/j.celrep.2022.110597>
- 13 Bogenhagen, D. F., Ostermeyer–Fay, A. G., Haley, J. D. & Garcia–Diaz, M. Kinetics and Mechanism of Mammalian Mitochondrial Ribosome Assembly. *Cell Rep* **22**, 1935–1944 (2018). <https://doi.org/10.1016/j.celrep.2018.01.066>
- 14 Vincent, T. *et al.* A genome-scale analysis of mRNAs targeting to plant mitochondria: upstream AUGs in 5' untranslated regions reduce mitochondrial

- association. *Plant J* **92**, 1132–1142 (2017). <https://doi.org:10.1111/tpj.13749>
- 15 Missbach, S. *et al.* 40S ribosome biogenesis co-factors are essential for gametophyte and embryo development. *Plos One* **8**, e54084 (2013). <https://doi.org:10.1371/journal.pone.0054084>
- 16 Li, P. C. *et al.* The AAA-ATPase MIDASIN 1 Functions in Ribosome Biogenesis and Is Essential for Embryo and Root Development. *Plant Physiol* **180**, 289–304 (2019). <https://doi.org:10.1104/pp.18.01225>
- 17 Li, K. *et al.* Arabidopsis NOTCHLESS plays an important role in root and embryo development. *Plant Signal Behav* **18**, 2245616 (2023). <https://doi.org:10.1080/15592324.2023.2245616>

Dear Editors,

We thank the editors and reviewers for the suggestions to improve the manuscript. Please find below our responses to the reviewer's comments.

REVIEWER COMMENTS

Reviewer #3 (Remarks to the Author):

Wang and colleagues have submitted a revised manuscript that thoroughly addresses most of the reviewers' comments. However, a few points still require attention:

1. Following on the previous comment (Rev#3 A) and given answer, it would be very useful to examine the interaction potential of RPL24 and OPNER/OAP using BiFC (since authors have that in hands), particularly given that BiFC is a robust system capable that may be reveal those transient interactions.

RE: We have tried the BiFC to investigate the interaction between RPL24C and the OPNR-OAP1-OAP2-CDC48D-CIP111 complex in tobacco leaves. However, we did not observe positive signals indicative of interaction. As previously discussed in the response to point 6 of reviewer 1, the interaction between the OPNR complex and RPL24C is transient and likely tightly regulated *in vivo*. The interaction may require conditions which are not recreated in the transient expression experiments in tobacco. For example, it is possible that the RPL24C should be in the pre-60S ribosome before it can act as a substrate for the OPNR complex *in vivo*, and that the interaction may also need additional trigger(s). Such added complexity may explain why the BiFC assay in tobacco leaves was not positive. We would also like to emphasize that for this assay to work it requires successful heterologous expression and assembly of the five proteins comprising the OPNR complex plus the RPL24C in the same epidermal tobacco cells. Thus, while we agree that it would be very useful to examine the interaction in a heterologous system it clearly represents a formidable experimental challenge.

Furthermore, we would also like to emphasize that the RPL24C was identified in the TurboID proximity labeling of CDC48D neighboring proteins as shown in Supplementary Fig. 4B. TurboID proximity labeling can catch transient interactions *in vivo* which fits with the proposed mode of action of the OPNR complex. We did not catch RPL24C in our Co-IP-MS experiments, in line with the fact that Co-IP does not detect transient interactions efficiently. And just about 2% of the His-OAP1-OAP2-OPNR trimer co-purified with GST-RPL24C in our pull-down *in vitro* assay as shown in Fig. 5C.

We would like to argue that the current results represent state-of-the-art methods in the field and considering that the heterologous assay of a multiprotein complex is likely

to require significant method development it should be considered for the future studies. Since both reviewer 1 and reviewer 2 are satisfied, and that the reviewer 3 writes “it would be useful to run BiFC assays”, we would kindly like the editors to reconsider this point.

2. I still think the representative FIB-SEM image, or link to the raw data should be provided.

RE: We have added the representative FIB-SEM images in Supplementary Fig. 4A-C.

3. The nomenclature should be consistent throughout the manuscript (Arabidopsis vs yeast). For example, gene/protein names such as Cell Division Cycle 48 D (CDC48D), Calmodulin-interacting protein 111 (CIP111), and Ribosomal-like protein 24 (RPL24) I guess it should follow the same formatting.

RE: We have changed yeast to *Saccharomyces cerevisiae*. And RIBOSOMAL PROTEIN L24C (RPL24C) was added in the abstract. And since the yeast nomenclature is different from plants, Ribosomal-like protein 24 was kept as Rlp24.

4. In Supplementary Figure S2F,G, the “%” symbol is unnecessary next to each number, as the figure title already states that values are expressed in percentages.

RE: “%” were removed in Supplementary Fig 2F, G.

5. In my opinion, the responses to the first and sixth comments from Reviewer 2 should be briefly summarized and incorporated into the main text of the manuscript, rather than remaining solely in the rebuttal.

RE: The responses to the first and sixth comments from Reviewer 2 were added on line 715-725 and lines 447-449 respectively.

6. Please review and correct the citation formatting and content throughout the manuscript—for instance, citation (46) and others may require revision.

RE: The citations were checked and revised.